# GET3D: A Generative Model of High Quality 3D Textured Shapes Learned from Images

**Jun Gao**[1,2,3]      **Tianchang Shen**[1,2,3]      **Zian Wang**[1,2,3]      **Wenzheng Chen**[1,2,3]

**Kangxue Yin**[1]      **Daiqing Li**[1]      **Or Litany**[1]      **Zan Gojcic**[1]      **Sanja Fidler**[1,2,3]

NVIDIA[1]      University of Toronto[2]      Vector Institute[3]

{jung, frshen, zianw, wenzchen, kangxuey, daiqingl, olitany, zgojcic, sfidler}@nvidia.com

## Abstract

As several industries are moving towards modeling massive 3D virtual worlds, the need for content creation tools that can scale in terms of the quantity, quality, and diversity of 3D content is becoming evident. In our work, we aim to train performant 3D generative models that synthesize textured meshes that can be directly consumed by 3D rendering engines, thus immediately usable in downstream applications. Prior works on 3D generative modeling either lack geometric details, are limited in the mesh topology they can produce, typically do not support textures, or utilize neural renderers in the synthesis process, which makes their use in common 3D software non-trivial. In this work, we introduce GET3D, a **G**enerative model that directly generates **E**xplicit **T**extured **3D** meshes with complex topology, rich geometric details, and high fidelity textures. We bridge recent success in the differentiable surface modeling, differentiable rendering as well as 2D Generative Adversarial Networks to train our model from 2D image collections. GET3D is able to generate high-quality 3D textured meshes, ranging from cars, chairs, animals, motorbikes and human characters to buildings, achieving significant improvements over previous methods. Our project page: https://nv-tlabs.github.io/GET3D

## 1   Introduction

Diverse, high-quality 3D content is becoming increasingly important for several industries, including gaming, robotics, architecture, and social platforms. However, manual creation of 3D assets is very time-consuming and requires specific technical knowledge as well as artistic modeling skills. One of the main challenges is thus scale – while one can find 3D models on 3D marketplaces such as Turbosquid [4] or Sketchfab [3], creating many 3D models to, say, populate a game or a movie with a crowd of characters that all look different still takes a significant amount of artist time.

To facilitate the content creation process and make it accessible to a variety of (novice) users, generative 3D networks that can produce high-quality and diverse 3D assets have recently become an active area of research [5, 13, 38, 41, 48, 59, 62, 55, 54, 60, 21]. However, to be practically useful for current real-world applications, 3D generative models should ideally fulfill the following requirements: **(a)** They should have the capacity to generate shapes with detailed geometry and arbitrary topology, **(b)** The output should be a textured mesh, which is a primary representation used by standard graphics software packages such as Blender [14] and Maya [1], and **(c)** We should be able to leverage 2D images for supervision, as they are more widely available than explicit 3D shapes.

Prior work on 3D generative modeling has focused on subsets of the above requirements, but no method to date fulfills all of them (Tab. 1). For example, methods that generate 3D point clouds [5,

36th Conference on Neural Information Processing Systems (NeurIPS 2022).

| Method | Application | Representation | Supervision | Textured mesh | Arbitrary topology |
|---|---|---|---|---|---|
| OccNet [38] | 3D generation | Implicit | 3D | ✗ | ✓ |
| PointFlow [59] | 3D generation | Point cloud | 3D | ✗ | ✓ |
| Texture3D [48] | 3D generation | Mesh | 2D | ✓ | ✗ |
| StyleNerf [23] | 3D-aware NV | Neural field | 2D | ✗ | ✓ |
| EG3D [8] | 3D-aware NV | Neural field | 2D | ✗ | ✓ |
| PiGAN [7] | 3D-aware NV | Neural field | 2D | ✗ | ✓ |
| GRAF [52] | 3D-aware NV | Neural field | 2D | ✗ | ✓ |
| Ours | 3D generation | Mesh | 2D | ✓ | ✓ |

Table 1: Comparison with prior works. (NV: Novel view synthesis.)

59, 62] typically do not produce textures and have to be converted to a mesh in post-processing. Methods generating voxels often lack geometric details and do not produce texture [57, 18, 25, 35]. Generative models based on neural fields [38, 13] focus on extracting geometry but disregard texture. Most of these also require explicit 3D supervision. Finally, methods that directly output textured 3D meshes [49, 48] typically require pre-defined shape templates and cannot generate shapes with complex topology and variable genus.

Recently, rapid progress in neural volume rendering [40] and 2D Generative Adversarial Networks (GANs) [31, 32, 30, 27, 47] has led to the rise of 3D-aware image synthesis [7, 52, 8, 44, 46, 23]. However, this line of work aims to synthesize multi-view consistent images using neural rendering in the synthesis process and does not guarantee that meaningful 3D shapes can be generated. While a mesh can potentially be obtained from the underlying neural field representation using the marching cube algorithm [34], extracting the corresponding texture is non-trivial.

In this work, we introduce a novel approach that aims to tackle all the requirements of a practically useful 3D generative model. Specifically, we propose GET3D, a **G**enerative model for 3D shapes that directly outputs **E**xplicit **T**extured **3D** meshes with high geometric and texture detail and arbitrary mesh topology. In the heart of our approach is a generative process that utilizes a differentiable *explicit* surface extraction method [55] and a differentiable rendering technique [42, 33]. The former enables us to directly optimize and output textured 3D meshes with arbitrary topology, while the latter allows us to train our model with 2D images, thus leveraging powerful and mature discriminators developed for 2D image synthesis. Since our model directly generates meshes and uses a highly efficient (differentiable) graphics renderer, we can easily scale up our model to train with image resolution as high as $1024 \times 1024$, allowing us to learn high-quality geometric and texture details.

We demonstrate state-of-the-art performance for unconditional 3D shape generation on multiple categories with complex geometry from ShapeNet [9], Turbosquid [4] and Renderpeople [2], such as chairs, motorbikes, cars, human characters, and buildings. With explicit mesh as output representation, GET3D is also very flexible and can easily be adapted to other tasks, including: **(a)** learning to generate decomposed material and view-dependent lighting effects using advanced differentiable rendering [12], without supervision, **(b)** text-guided 3D shape generation using CLIP [51] embedding.

## 2 Related Work

We review recent advances in 3D generative models for geometry and appearance, as well as 3D-aware generative image synthesis.

**3D Generative Models**   In recent years, 2D generative models have achieved photorealistic quality in high-resolution image synthesis [31, 32, 30, 47, 27, 17, 15]. This progress has also inspired research in 3D content generation. Early approaches aimed to directly extend the 2D CNN generators to 3D voxel grids [57, 18, 25, 35, 56], but the high memory footprint and computational complexity of 3D convolutions hinder the generation process at high resolution. As an alternative, other works have explored point cloud [5, 59, 62, 41], implicit [38, 13], or octree [28] representations. However, these works focus mainly on generating geometry and disregard appearance. Their output representations also need to be post-processed to make them compatible with standard graphics engines.

More similar to our work, Textured3DGAN [49, 48] and DIBR [11] generate textured 3D meshes, but they formulate the generation as a deformation of a template mesh, which prevents them from generating complex topology or shapes with varying genus, which our method can do. PolyGen [43] and SurfGen [36] can produce meshes with arbitrary topology, but do not synthesize textures.

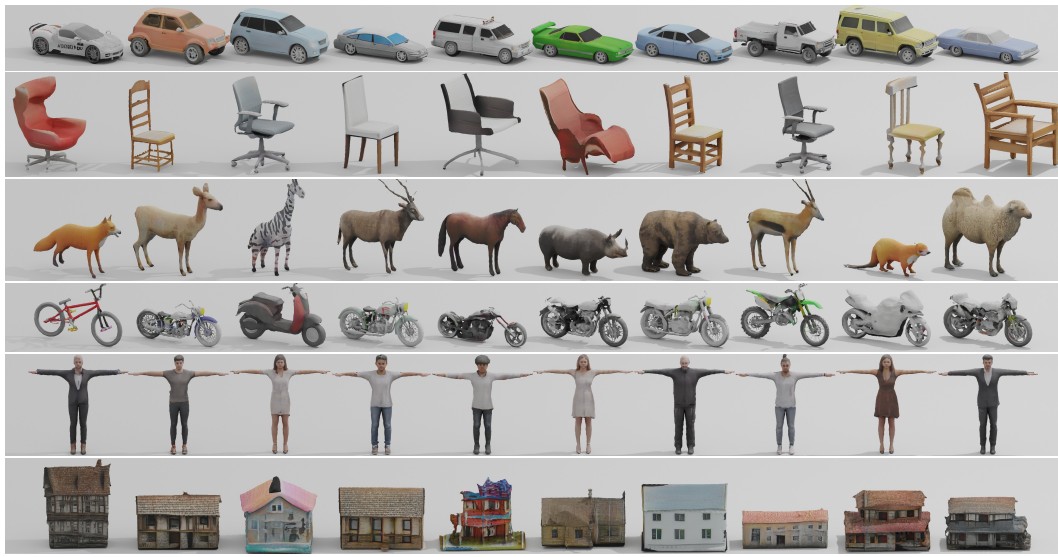

Figure 1: We export our **generated shapes** and visualize them in Blender. GET3D is able to generate diverse shapes with arbitrary topology, high quality geometry, and texture.

**3D-Aware Generative Image Synthesis**    Inspired by the success of neural volume rendering [40] and implicit representations [38, 13], recent work started tackling the problem of 3D-aware image synthesis [7, 52, 44, 24, 23, 63, 8, 46, 53, 58]. However, neural volume rendering networks are typically slow to query, leading to long training times [7, 52], and generate images of limited resolution. GIRAFFE [44] and StyleNerf [23] improve the training and rendering efficiency by performing neural rendering at a lower resolution and then upsampling the results with a 2D CNN. However, the performance gain comes at the cost of a reduced multi-view consistency. By utilizing a dual discriminator, EG3D [8] can partially mitigate this problem. Nevertheless, extracting a textured surface from methods that are based on neural rendering is a non-trivial endeavor. In contrast, GET3D directly outputs textured 3D meshes that can be readily used in standard graphics engines.

# 3    Method

We now present our GET3D framework for synthesizing textured 3D shapes. Our generation process is split into two parts: a geometry branch, which differentiably outputs a surface mesh of arbitrary topology, and a texture branch that produces a texture field that can be queried at the surface points to produce colors. The latter can be extended to other surface properties such as for example materials (Sec. 4.3.1). During training, an efficient differentiable rasterizer is utilized to render the resulting textured mesh into 2D high-resolution images. The entire process is differentiable, allowing for adversarial training from images (with masks indicating an object of interest) by propagating the gradients from the 2D discriminator to both generator branches. Our model is illustrated in Fig. 2. In the following, we first introduce our 3D generator in Sec 3.1, before proceeding to the differentiable rendering and loss functions in Sec 3.2.

## 3.1    Generative Model of 3D Textured Meshes

We aim to learn a 3D generator $M, E = G(\mathbf{z})$ to map a sample from a Gaussian distribution $\mathbf{z} \in \mathcal{N}(0, \mathbf{I})$ to a mesh $M$ with texture $E$.

Since the same geometry can have different textures, and the same texture can be applied to different geometries, we sample two random input vectors $\mathbf{z}_1 \in \mathbb{R}^{512}$ and $\mathbf{z}_2 \in \mathbb{R}^{512}$. Following StyleGAN [31, 32, 30], we then use non-linear mapping networks $f_{\text{geo}}$ and $f_{\text{tex}}$ to map $\mathbf{z}_1$ and $\mathbf{z}_2$ to intermediate latent vectors $\mathbf{w}_1 = f_{\text{geo}}(\mathbf{z}_1)$ and $\mathbf{w}_2 = f_{\text{tex}}(\mathbf{z}_2)$ which are further used to produce *styles* that control the generation of 3D shapes and texture, respectively. We formally introduce the generator for geometry in Sec. 3.1.1 and the texture generator in Sec. 3.1.2.

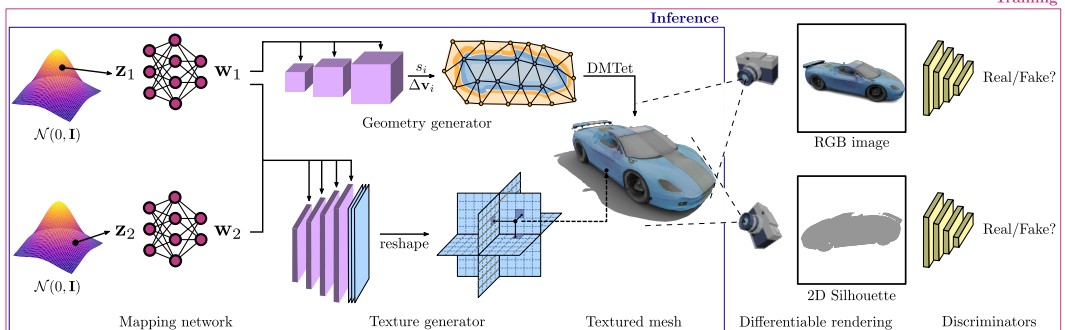

Inference

$\mathcal{N}(0, \mathbf{I})$

Mapping network     Texture generator     Textured mesh     Differentiable rendering     Discriminators

Figure 2: Overview of **GET3D:** We generate a 3D SDF and a texture field via two latent codes. We utilize DMTet [55] to extract a 3D surface mesh from the SDF, and query the texture field at surface points to get colors. We train with adversarial losses defined on 2D images. In particular, we use a rasterization-based differentiable renderer [33] to obtain RGB images and silhouettes. We utilize two 2D discriminators, each on RGB image, and silhouette, respectively, to classify whether the inputs are real or fake. The whole model is end-to-end trainable. Note that we additionally provide an improved version of our Generator in Appendix Sec. A.5 and Fig. C.

### 3.1.1 Geometry Generator

We design our geometry generator to incorporate DMTet [55], a recently proposed differentiable surface representation. DMTet represents geometry as a signed distance field (SDF) defined on a deformable tetrahedral grid [20, 22], from which the surface can be differentiably recovered through marching tetrahedra [16]. Deforming the grid by moving its vertices results in a better utilization of its resolution. By adopting DMTet for surface extraction, we can produce explicit meshes with arbitrary topology and genus. We next provide a brief summary of DMTet and refer the reader to the original paper for further details.

Let $(V_T, T)$ denote the full 3D space that the object lies in, where $V_T$ are the vertices in the tetrahedral grid $T$. Each tetrahedron $T_k \in T$ is defined using four vertices $\{\mathbf{v}_{a_k}, \mathbf{v}_{b_k}, \mathbf{v}_{c_k}, \mathbf{v}_{d_k}\}$, with $k \in \{1, \ldots, K\}$, where $K$ is the total number of tetrahedra, and $\mathbf{v}_{i_k} \in V_T, \mathbf{v}_{i_k} \in \mathbb{R}^3$. In addition to its 3D coordinates, each vertex $\mathbf{v}_i$ contains the SDF value $s_i \in \mathbb{R}$ and the deformation $\Delta \mathbf{v}_i \in \mathbb{R}^3$ of the vertex from its initial canonical coordinate. This representation allows recovering the explicit mesh through differentiable marching tetrahedra [55], where SDF values in continuous space are computed by a barycentric interpolation of their value $s_i$ on the deformed vertices $\mathbf{v}'_i = \mathbf{v}_i + \Delta \mathbf{v}_i$.

**Network Architecture**    We map $\mathbf{w}_1 \in \mathbb{R}^{512}$ to SDF values and deformations at each vertex $\mathbf{v}_i$ through a series of conditional 3D convolutional and fully connected layers. Specifically, we first use 3D convolutional layers to generate a feature volume conditioned on $\mathbf{w}_1$. We then query the feature at each vertex $\mathbf{v}_i \in V_T$ using trilinear interpolation and feed it into MLPs that outputs the SDF value $s_i$ and the deformation $\Delta \mathbf{v}_i$. In cases where modeling at a high-resolution is required (e.g. motorbike with thin structures in the wheels), we further use volume subdivision following [55].

**Differentiable Mesh Extraction**    After obtaining $s_i$ and $\Delta \mathbf{v}_i$ for all the vertices, we use the differentiable marching tetrahedra algorithm to extract the explicit mesh. Marching tetrahedra determines the surface topology within each tetrahedron based on the signs of $s_i$. In particular, a mesh face is extracted when $\text{sign}(s_i) \neq \text{sign}(s_j)$, where $i, j$ denotes the indices of vertices in the edge of tetrahedron, and the vertices $\mathbf{m}_{i,j}$ of that face are determined by a linear interpolation as $\mathbf{m}_{i,j} = \frac{\mathbf{v}'_i s_j - \mathbf{v}'_j s_i}{s_j - s_i}$. Note that the above equation is only evaluated when $s_i \neq s_j$, thus it is differentiable, and the gradient from $\mathbf{m}_{i,j}$ can be back-propagated into the SDF values $s_i$ and deformations $\Delta \mathbf{v}_i$. With this representation, the shapes with arbitrary topology can easily be generated by predicting different signs of $s_i$.

### 3.1.2 Texture Generator

Directly generating a texture map consistent with the output mesh is not trivial, as the generated shape can have an arbitrary genus and topology. We thus parameterize the texture as a texture field [45].

Specifically, we model the texture field with a function $f_t$ that maps the 3D location of a surface point $\mathbf{p} \in \mathbb{R}^3$, conditioned on the $\mathbf{w}_2$, to the RGB color $\mathbf{c} \in \mathbb{R}^3$ at that location. Since the texture field depends on geometry, we additionally condition this mapping on the geometry latent code $\mathbf{w}_1$, such that $\mathbf{c} = f_t(\mathbf{p}, \mathbf{w}_1 \oplus \mathbf{w}_2)$, where $\oplus$ denotes concatenation.

**Network Architecture**  We represent our texture field using a tri-plane representation, which is efficient and expressive in reconstructing 3D objects [50] and generating 3D-aware images [8] . Specifically, we follow [8, 32] and use a conditional 2D convolutional neural network to map the latent code $\mathbf{w}_1 \oplus \mathbf{w}_2$ to three axis-aligned orthogonal feature planes of size $N \times N \times (C \times 3)$, where $N = 256$ denotes the spatial resolution and $C = 32$ the number of channels.

Given the feature planes, the feature vector $\mathbf{f}^t \in \mathbb{R}^{32}$ of a surface point $\mathbf{p}$ can be recovered as $\mathbf{f}^t = \sum_e \rho(\pi_e(\mathbf{p}))$, where $\pi_e(\mathbf{p})$ is the projection of the point $\mathbf{p}$ to the feature plane $e$ and $\rho(\cdot)$ denotes bilinear interpolation of the features. An additional fully connected layer is then used to map the aggregated feature vector $\mathbf{f}^t$ to the RGB color $\mathbf{c}$. Note that, different from other works on 3D-aware image synthesis [8, 23, 7, 52] that also use a neural field representation, we only need to sample the texture field at the locations of the surface points (as opposed to dense samples along a ray). This greatly reduces the computational complexity for rendering high-resolution images and guarantees to generate multi-view consistent images by construction.

## 3.2  Differentiable Rendering and Training

In order to supervise our model during training, we draw inspiration from Nvdiffrec [42] that performs multi-view 3D object reconstruction by utilizing a differentiable renderer. Specifically, we render the extracted 3D mesh and the texture field into 2D images using a differentiable renderer [33], and supervise our network with a 2D discriminator, which tries to distinguish the image from a real object or rendered from the generated object.

**Differentiable Rendering**  We assume that the camera distribution $\mathcal{C}$ that was used to acquire the images in the dataset is known. To render the generated shapes, we randomly sample a camera $c$ from $\mathcal{C}$, and utilize a highly-optimized differentiable rasterizer Nvdiffrast [33] to render the 3D mesh into a 2D silhouette as well as an image where each pixel contains the coordinates of the corresponding 3D point on the mesh surface. These coordinates are further used to query the texture field to obtain the RGB values. Since we operate directly on the extracted mesh, we can render high-resolution images with high efficiency, allowing our model to be trained with image resolution as high as $1024 \times 1024$.

**Discriminator & Objective**  We train our model using an adversarial objective. We adopt the discriminator architecture from StyleGAN [31], and use the same non-saturating GAN objective with R1 regularization [37]. We empirically find that using two separate discriminators, one for RGB images and another one for silhouettes, yields better results than a single discriminator operating on both. Let $D_x$ denote the discriminator, where $x$ can either be an RGB image or a silhouette. The adversarial objective is then be defined as follows:

$$L(D_x, G) = \mathbb{E}_{\mathbf{z} \in \mathcal{N}, c \in \mathcal{C}}[g(D_x(R(G(\mathbf{z}), c)))] + \mathbb{E}_{I_x \in p_x}[g(-D_x(I_x)) + \lambda ||\nabla D_x(I_x)||_2^2], \tag{1}$$

where $g(u)$ is defined as $g(u) = -\log(1 + \exp(-u))$, $p_x$ is the distribution of real images, $R$ denotes rendering, and $\lambda$ is a hyperparameter. Since $R$ is differentiable, the gradients can be backpropagated from 2D images to our 3D generators.

**Regularization**  To remove internal floating faces that are not visible in any of the views, we further regularize the geometry generator with a cross-entropy loss defined between the SDF values of the neighboring vertices [42]:

$$L_{\text{reg}} = \sum_{i,j \in \mathbb{S}_e} H\left(\sigma(s_i), \text{sign}(s_j)\right) + H\left(\sigma(s_j), \text{sign}(s_i)\right), \tag{2}$$

where $H$ denotes binary cross-entropy loss and $\sigma$ denotes the sigmoid function. The sum in Eq. 2 is defined over the set of unique edges $\mathbb{S}_e$ in the tetrahedral grid, for which $\text{sign}(s_i) \neq \text{sign}(s_j)$.

The overall loss function is then defined as:

$$L = L(D_{\text{rgb}}, G) + L(D_{\text{mask}}, G) + \mu L_{\text{reg}}, \tag{3}$$

where $\mu$ is a hyperparameter that controls the level of regularization.

| Category | Method | COV (%, ↑) | | MMD (↓) | | FID (↓) | |
|---|---|---|---|---|---|---|---|
| | | LFD | CD | LFD | CD | Ori | 3D |
| Car | PointFlow [59] | 51.91 | 57.16 | 1971 | 0.82 | - | - |
| | OccNet [38] | 27.29 | 42.63 | 1717 | **0.61** | - | - |
| | Pi-GAN [7] | 0.82 | 0.55 | 6626 | 25.54 | 52.82 | 104.29 |
| | GRAF [52] | 1.57 | 1.57 | 6012 | 10.63 | 49.95 | 52.85 |
| | EG3D [8] | 60.16 | 49.52 | 1527 | 0.72 | 15.52 | 21.89 |
| | Ours | **66.78** | **58.39** | **1491** | 0.71 | **10.25** | **10.25** |
| | Ours+Subdiv. | 62.48 | 55.93 | 1553 | 0.72 | 12.14 | 12.14 |
| | Ours (improved $G$) | 59.00 | 47.95 | 1473 | 0.81 | 10.60 | 10.60 |
| Chair | PointFlow [59] | 49.58 | **71.87** | 3755 | **3.03** | - | - |
| | OccNet [38] | 61.10 | 67.13 | 3494 | 3.98 | - | - |
| | Pi-GAN [7] | 53.76 | 39.65 | 4092 | 6.65 | 65.70 | 120.53 |
| | GRAF [52] | 50.23 | 39.28 | 4055 | 6.80 | 43.82 | 61.63 |
| | EG3D [8] | 58.31 | 50.14 | 3444 | 4.72 | 38.87 | 46.06 |
| | Ours | 69.08 | 69.91 | 3167 | 3.72 | 23.28 | 23.28 |
| | Ours+Subdiv. | **71.59** | 70.84 | **3163** | 3.95 | **23.17** | **23.17** |
| | Ours (improved $G$) | 71.96 | 71.96 | 3125 | 3.96 | 22.41 | 22.41 |

| Category | Method | COV (%, ↑) | | MMD (↓) | | FID (↓) | |
|---|---|---|---|---|---|---|---|
| | | LFD | CD | LFD | CD | Ori | 3D |
| Mbike | PointFlow [59] | 50.68 | 63.01 | 4023 | **1.38** | - | - |
| | OccNet [38] | 30.14 | 47.95 | 4551 | 2.04 | - | - |
| | Pi-GAN [7] | 2.74 | 6.85 | 8864 | 21.08 | 72.67 | 131.38 |
| | GRAF [52] | 43.84 | 50.68 | 4528 | 2.40 | 83.20 | 113.39 |
| | EG3D [8] | 38.36 | 34.25 | 4199 | 2.21 | 66.38 | 89.97 |
| | Ours | 67.12 | **67.12** | 3631 | 1.72 | 65.60 | 65.60 |
| | Ours+Subdiv. | 63.01 | 61.64 | **3440** | 1.79 | 54.12 | 54.12 |
| | Ours (improved $G$) | **69.86** | 65.75 | 3393 | 1.79 | **48.90** | **48.90** |
| Animal | PointFlow [59] | 42.70 | 74.16 | 4885 | **1.68** | - | - |
| | OccNet [38] | 56.18 | 75.28 | 4418 | 2.39 | - | - |
| | Pi-GAN [7] | 31.46 | 30.34 | 6084 | 8.37 | 36.26 | 150.86 |
| | GRAF [52] | 60.67 | 61.80 | 5083 | 4.81 | 42.07 | 52.48 |
| | EG3D [8] | 74.16 | 58.43 | 4889 | 3.42 | 40.03 | 83.47 |
| | Ours | **79.77** | 78.65 | **3798** | 2.02 | **28.33** | **28.33** |
| | Ours+Subdiv. | 66.29 | 74.16 | 3864 | 2.03 | 28.49 | 28.49 |
| | Ours (improved $G$) | 74.16 | **82.02** | 3767 | 1.97 | 27.18 | 27.18 |

Table 2: **Quantitative evaluation of generation results**: ↑: the higher the better, ↓: the lower the better. The best scores are highlighted in bold. MMD-CD scores are multiplied by $10^3$. The results of *Ours (improved G)* were obtained after the review process by improving the design of the generator network architecture $G$ (see Appendix A.5 for more details).

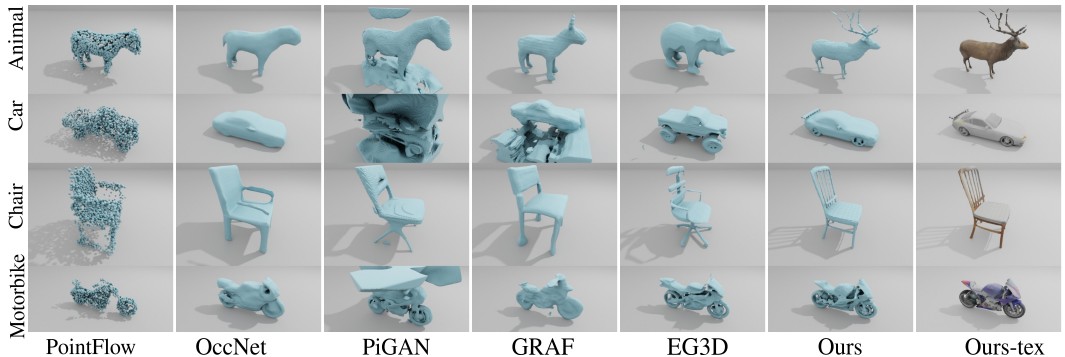

Figure 3: Qualitative comparison of GET3D to the baseline methods in terms of extracted 3D geometry. GET3D is able to generate shapes with much higher geometric detail across all categories.

# 4 Experiments

We conduct extensive experiments to evaluate our model. We first compare the quality of the 3D textured meshes generated by GET3D to the existing methods using the ShapeNet [9] and Turbosquid [4] datasets. Next, we ablate our design choices in Sec. 4.2. Finally, we demonstrate the flexibility of GET3D by adapting it to downstream applications in Sec. 4.3. Additional experimental results and implementation details are provided in Appendix.

## 4.1 Experiments on Synthetic Datasets

**Datasets** For evaluation on ShapeNet [9], we use three categories with complex geometry – *Car*, *Chair*, and *Motorbike*, which contain 7497, 6778, and 337 shapes, respectively. We randomly split each category into training (70%), validation (10 %), and test (20 %), and further remove from the test set shapes that have duplicates in the training set. To render the training data, we randomly sample camera poses from the upper hemisphere of each shape. For the *Car* and *Chair* categories, we use 24 random views, while for *Motorbike* we use 100 views due to less number of shapes. As models in ShapeNet only have simple textures, we also evaluate GET3D on an *Animal* dataset (442 shapes) collected from TurboSquid [4], where textures are more detailed and we split it into training, validation and test as defined above. Finally, to demonstrate the versatility of GET3D, we also provide qualitative results on the *House* dataset collected from Turbosquid (563 shapes), and *Human Body* dataset from Renderpeople [2] (500 shapes). We train a separate model on each category.

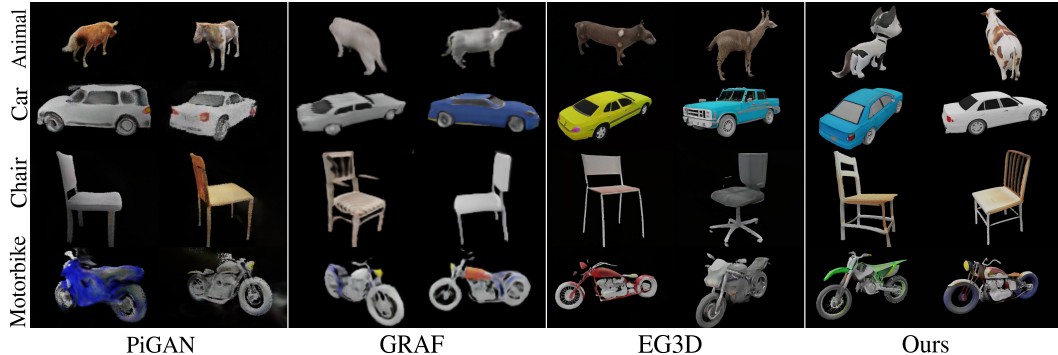

Figure 4: Qualitative comparison of GET3D to the baseline methods in terms of generated 2D images. GET3D generates sharp textures with high level of detail.

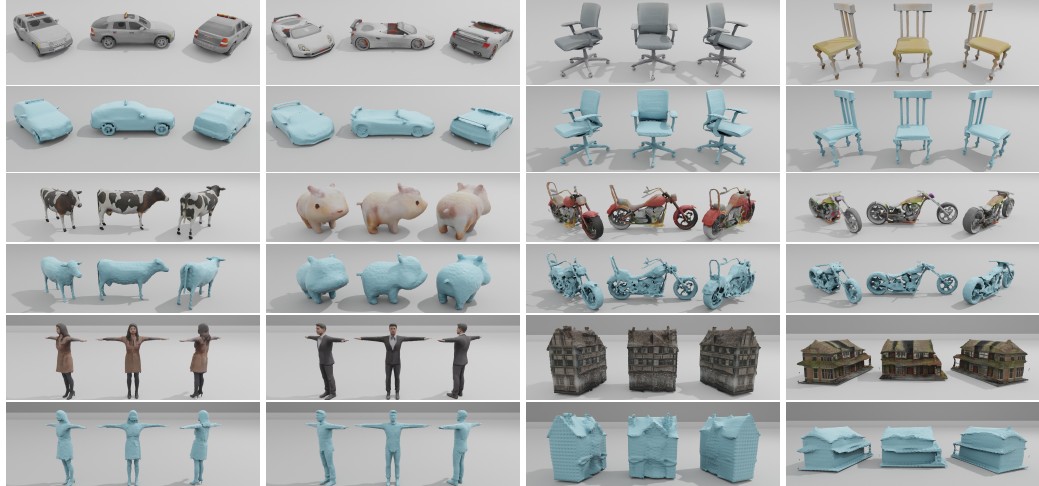

Figure 5: **Shapes generated by GET3D rendered in Blender.** GET3D generates high-quality shapes with diverse texture, high-quality geometry, and complex topology. Zoom-in for details.

**Baselines** We compare GET3D to two groups of works: **1)** 3D generative models that rely on 3D supervision: PointFlow [59] and OccNet [38]. Note that these methods only generate geometry without texture. **2)** 3D-aware image generation methods: GRAF [52], PiGAN [7], and EG3D [8].

**Metrics** To evaluate the quality of our synthesis, we consider both the geometry and texture of the generated shapes. For geometry, we adopt metrics from [5] and use both Chamfer Distance (CD) and Light Field Distance [10] (LFD) to compute the Coverage score and Minimum Matching Distance. For OccNet [38], GRAF [52], PiGAN [7] and EG3D [8], we use marching cubes to extract the underlying geometry. For PointFlow [59], we use Poisson surface reconstruction to convert a point cloud into a mesh when evaluating LFD. To evaluate texture quality, we adopt the FID [26] metric commonly used to evaluate image synthesis. In particular, for each category, we render the test shapes into 2D images, and also render the generated 3D shapes from each model into 50k images using the same camera distribution. We then compute FID on the two image sets. As the baselines from 3D-aware image synthesis [52, 7, 8] do not directly output textured meshes, we compute FID score in two ways: (**i**) we use their neural volume rendering to obtain 2D images, which we refer to as FID-Ori, and (**ii**) we extract the mesh from their neural field representation using marching cubes, render it, and then use the 3D location of each pixel to query the network to obtain the RGB values. We refer to this score, that is more aware of the actual 3D shape, as FID-3D. Further details on the evaluation metrics are available in the Appendix B.3.

**Experimental Results** We provide quantitative results in Table. 2 and qualitative examples in Fig. 3 and Fig. 4. Additional results are available in the supplementary video. Compared to OccNet [38] that

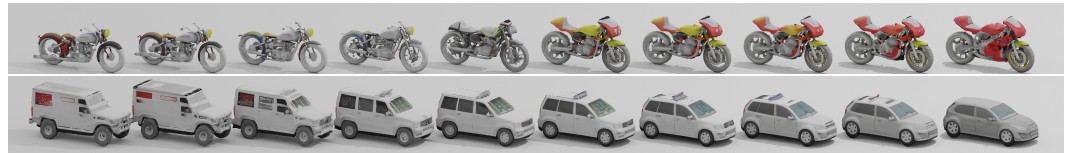

Figure 6: **Shape interpolation.** We interpolate both geometry and texture latent codes from left to right.

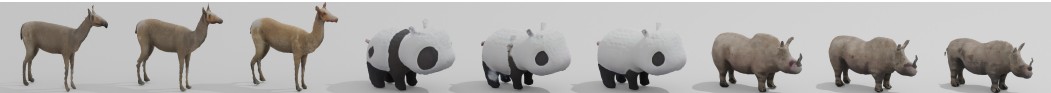

Figure 7: **Shape variation.** We locally perturb each latent code to generate different shapes.

uses 3D supervision during training, GET3D achieves better performance in terms of both diversity (COV) and quality (MMD), and our generated shapes have more geometric details. PointFlow [59] outperforms GET3D in terms of MMD on CD, while GET3D is better in MMD on LFG. We hypothesize that this is because PointFlow directly optimizes on point locations, which favours CD. GET3D also performs favourably when compared to 3D-aware image synthesis methods, we achieve significant improvements over PiGAN [7] and GRAF [52] in terms of all metrics on all datasets. Our generated shapes also contain more detailed geometry and texture. Compared with recent work EG3D [8]. We achieve comparable performance on generating 2D images (FID-ori), while we significantly improve on 3D shape synthesis in terms of FID-3D, which demonstrates the effectiveness of our model on learning actual 3D geometry and texture.

Since we synthesize textured meshes, we can export our shapes into Blender[1]. We show rendering results in Fig. 1 and 5. GET3D is able to generate shapes with diverse and high quality geometry and topology, very thin structures (motorbikes), as well as complex textures on cars, animals, and houses.

**Shape Interpolation** GET3D also enables shape interpolation, which can be useful for editing purposes. We explore the latent space of GET3D in Fig. 6, where we interpolate the latent codes to generate each shape from left to right. GET3D is able to faithfully generate a smooth and meaningful transition from one shape to another. We further explore the local latent space by slightly perturbing the latent codes to a random direction. GET3D produces novel and diverse shapes when applying local editing in the latent space (Fig. 7).

## 4.2 Ablations

We ablate our model in two ways: **1)** w/ and w/o volume subdivision, **2)** training using different image resolutions. Further ablations are provided in the Appendix C.3.

**Ablation of Volume Subdivision** As shown in Tbl. 2, volume subdivision significantly improves the performance on classes with thin structures (e.g., motorbikes), while not getting gains on other classes. We hypothesize that the initial tetrahedral resolution is already sufficient to capture the detailed geometry on Chairs and Cars, and hence the subdivision cannot provide further improvements.

**Ablating Different Image Resolutions** We ablate the effect of the training image resolution in Tbl. 3. As expected, increased image resolution

| Class | Img Res | COV (%, ↑) | | MMD (↓) | | FID (↓) |
|---|---|---|---|---|---|---|
| | | LFD | CD | LFD | CD | |
| Car | $128^2$ | 9.28 | 8.25 | 2224 | 1.30 | 39.21 |
| | $512^2$ | 52.32 | 44.13 | 1593 | 0.80 | 13.19 |
| | $1024^2$ | **66.78** | **58.39** | **1491** | **0.71** | **10.25** |
| Chair | $128^2$ | 38.25 | 33.98 | 3886 | 5.90 | 43.04 |
| | $512^2$ | 68.80 | **69.92** | **3149** | 3.90 | 30.16 |
| | $1024^2$ | **69.08** | 67.87 | 3167 | **3.74** | **23.28** |
| Mbike | $512^2$ | **68.49** | **65.75** | 3421 | 1.74 | 74.04 |
| | $1024^2$ | 67.12 | 64.38 | 3631 | **1.73** | **65.60** |
| Animal | $512^2$ | 77.53 | **78.65** | 3828 | **2.01** | 29.75 |
| | $1024^2$ | **79.78** | **78.65** | **3798** | 2.03 | **28.33** |

Table 3: **Ablating the image resolution.** ↑: higher is better, ↓: lower is better.

[1]We use xatlas [61] to get texture coordinates for the extracted mesh, from where we can warp our 3D mesh into a 2D plane and obtain the corresponding 3D location on the mesh surface for any position on the 2D plane. We then discretize the 2D plane into an image, and for each pixel, we query the texture field using corresponding 3D location to obtain the RGB color to get the texture map.

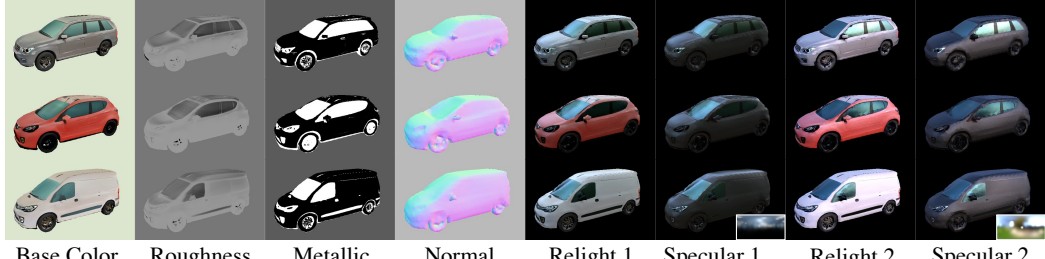

| Base Color | Roughness | Metallic | Normal | Relight 1 | Specular 1 | Relight 2 | Specular 2 |

Figure 8: **Material generation and relighting.** Despite being unsupervised, our model generates reasonable material properties, and can be realistically rendered with real-world HDR panoramas (bottom right). Normals are computed from the generated mesh. Note how specular effects change under two different lighting conditions.

improves the performance in terms of FID and shape quality, as the network can see more details, which are often not available in the low-resolution images. This corroborates the importance of training with higher image resolution, which are often hard to make use of for implicit-based methods.

### 4.3 Applications

#### 4.3.1 Material Generation for View-dependent Lighting Effects

GET3D can easily be extended to also generate surface materials that are directly usable in modern graphics engines. In particular, we follow the widely used Disney BRDF [6, 29] and describe the materials in terms of the base color ($\mathbb{R}^3$), metallic ($\mathbb{R}$), and roughness ($\mathbb{R}$) properties. As a result, we repurpose our texture generator to now output a 5-channel reflectance field (instead of only RGB). To accommodate differentiable rendering of materials, we adopt an efficient spherical Gaussian (SG) based deferred rendering pipeline [12]. Specifically, we rasterize the reflectance field into a G-buffer, and randomly sample an HDR image from a set of real-world outdoor HDR panoramas $\mathcal{S}_{\text{light}} = \{L_{SG}\}_K$, where $L_{SG} \in \mathbb{R}^{32 \times 7}$ is obtained by fitting 32 SG lobes to each panorama. The SG renderer [12] then uses the camera $c$ to render an RGB image with view-dependent lighting effects, which we feed into the discriminator during training. Note that GET3D does not require material supervision during training and learns to generate decomposed materials in an unsupervised manner.

We provide qualitative results of generated surface materials in Fig. 8. Despite unsupervised, GET3D discovers interesting material decomposition, e.g., the windows are correctly predicted with a smaller roughness value to be more glossy than the car's body, and the car's body is discovered as more dielectric while the window is more metallic. Generated materials enable us to produce realistic relighting results, which can account for complex specular effects under different lighting conditions.

#### 4.3.2 Text-Guided 3D Synthesis

Similar to image GANs, GET3D also supports text-guided 3D content synthesis by fine-tuning a pre-trained model under the guidance of CLIP [51]. Note that our final synthesis result is a textured 3D mesh. To this end, we follow the dual-generator design from styleGAN-NADA [19], where a trainable copy $G_t$ and a frozen copy $G_f$ of the pre-trained generator are adopted. During optimization $G_t$ and $G_f$ both render images from 16 random camera views. Given a text query, we sample 500 pairs of noise vectors $\mathbf{z}_1$ and $\mathbf{z}_2$. For each sample, we optimize the parameters of $G_t$ to minimize the directional CLIP loss [19] (the source text labels are "car", "animal" and "house" for the corresponding categories), and select the samples with minimal loss. To accelerate this process, we first run a small number of optimization steps for the 500 samples, then choose the top 50 samples with the lowest losses, and run the optimization for 300 steps. The results and comparison against a SOTA text-driven mesh stylization method, Text2Mesh [39], are provided in Fig. 9. Note that, [39] requires a mesh of the shape as an input to the method. We provide our generated meshes from the frozen generator as input meshes to it. Since it needs mesh vertices to be dense to synthesize surface details with vertex displacements, we further subdivide the input meshes with mid-point subdivision to make sure each mesh has 50k-150k vertices on average.

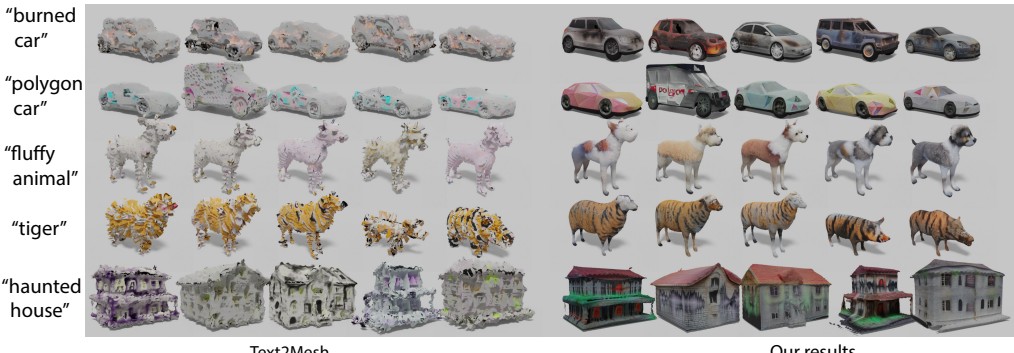

Text2Mesh · · · · · · · · · · · · · · · · · · · · · · · · · · · · · · · Our results

Figure 9: **Text-guided 3D synthesis.** Note that Text2Mesh [39] requires 3D mesh geometry as input. To fulfil the requirement, we provide our generated geometry as its input mesh.

## 5 Conclusion

We introduced GET3D, a novel 3D generative model that is able to synthesize high-quality 3D textured meshes with arbitrary topology. GET3D is trained using only 2D images as supervision. We experimentally demonstrated significant improvements on generating 3D shapes over previous state-of-the-art methods on multiple categories. We hope that this work brings us one step closer to democratizing 3D content creation using A.I..

**Limitations**  While GET3D makes a significant step towards a practically useful 3D generative model of 3D textured shapes, it still has some limitations. In particular, we still rely on 2D silhouettes as well as the knowledge of camera distribution during training. As a consequence, GET3D was currently only evaluated on synthetic data. A promising extension could use the advances in instance segmentation and camera pose estimation to mitigate this issue and extend GET3D to real-world data. GET3D is also trained per-category; extending it to multiple categories in the future, could help us better represent the inter-category diversity.

**Broader Impact**  We proposed a novel 3D generative model that generates 3D textured meshes, which can be readily imported into current graphics engines. Our model is able to generate shapes with arbitrary topology, high quality textures and rich geometric details, paving the path for democratizing A.I. tool for 3D content creation. As all machine learning models, GET3D is also prone to biases introduced in the training data. Therefore, an abundance of caution should be applied when dealing with sensitive applications, such as generating 3D human bodies, as GET3D is not tailored for these applications. We do not recommend using GET3D if privacy or erroneous recognition could lead to potential misuse or any other harmful applications. Instead, we do encourage practitioners to carefully inspect and de-bias the datasets before training our model to depict a fair and wide distribution of possible skin tones, races or gender identities.

## 6 Disclosure of Funding

This work was funded by NVIDIA. Jun Gao, Tianchang Shen, Zian Wang and Wenzheng Chen acknowledge additional revenue in the form of student scholarships from University of Toronto and the Vector Institute, which are not in direct support of this work.

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
