# Appendix
# GET3D: A Generative Model of High Quality 3D Textured Shapes Learned from Images

**Jun Gao**[1,2,3]    **Tianchang Shen**[1,2,3]    **Zian Wang**[1,2,3]    **Wenzheng Chen**[1,2,3]

**Kangxue Yin**[1]    **Daiqing Li**[1]    **Or Litany**[1]    **Zan Gojcic**[1]    **Sanja Fidler**[1,2,3]

NVIDIA[1]    University of Toronto[2]    Vector Institute[3]

{jung, frshen, zianw, wenzchen, kangxuey, daiqingl, olitany, zgojcic, sfidler}@nvidia.com

In this Appendix, we first provide detailed description of the GET3D network architecture (Sec. A.1-A.4) along with the training procedure and hyperparameters (Sec. A.6). We then describe the datasets (Sec. B.1), baselines (Sec. B.2), and evaluation metrics (Sec. B.3). Additional qualitative results, ablation studies, robustness analysis, and results on the real dataset are available in Sec. C. Details and additional results of the material generation for view-dependent lighting effects are provided in Sec. D. Sec E contains more information about the text-guided shape generation experiments as well as more additional qualitative results. The readers are also kindly referred to the accompanying video (*demo.mp4*) that includes 360-degree renderings of our results (more than 400 generated shapes for each category), detailed zoom-ins, interpolations, material generation, and shapes generated with text-guidance.

## A  Details of Our Model

In the main paper, we have provided a high level description of GET3D. Here, we provide the implementation details that were omitted due to the lack of space. Please consult the Figure B and Figure 2 in the main paper for more context. Source code is available at our project webpage

### A.1  Mapping Network

Following StyleGAN [18, 19], our mapping networks $f_{\text{geo}}$ and $f_{\text{tex}}$ are 8-layer MLPs in which each fully-connected layer has 512 hidden dimensions and a leaky-ReLU activation (Figure B). The mapping networks are used to map the randomly sampled noise vectors $\mathbf{z}_1 \in \mathbb{R}^{512}$ and $\mathbf{z}_2 \in \mathbb{R}^{512}$ to the latent vectors $\mathbf{w}_1 \in \mathbb{R}^{512}$ and $\mathbf{w}_2 \in \mathbb{R}^{512}$ as $\mathbf{w}_1 = f_{\text{geo}}(\mathbf{z}_1)$ and $\mathbf{w}_2 = f_{\text{tex}}(\mathbf{z}_2)$.

### A.2  Geometry Generator

The geometry generator of GET3D starts from a randomly initialized feature volume $\mathbf{F}_{\text{geo}} \in \mathbb{R}^{4 \times 4 \times 4 \times 256}$ that is shared across the generated shapes, and is learned during training. Through a series of four modulated 3D convolution blocks (*ModBlock3D* in Figure B), the initial volume is up-sampled to a feature volume $\mathbf{F}'_{\text{geo}} \in \mathbb{R}^{32 \times 32 \times 32 \times 64}$ that is conditioned on $\mathbf{w}_1$. Specifically, in each *ModBlock3D*, the input feature volume is first upsampled by a factor of two using trilinear interpolation. It is then passed through a small 3D ResNet, where the residual path uses a 3D convolutional layer with kernel size 1x1x1, and the main path applies two *conditional* 3D convolutional layers with kernel size 3x3x3. To perform the conditioning, we follow StyleGAN2 [19] and first map the latent vector $\mathbf{w}_1$ to *style* $\mathbf{h}$ through a learned affine transformation ($\mathbf{A}$ in Figure B). The *style* $\mathbf{h}$ is then used to modulate ($\mathbf{M}$) and demodulate ($\mathbf{D}$) the weights of the convolutional layers as:

36th Conference on Neural Information Processing Systems (NeurIPS 2022).

$$\mathbf{M}: \theta'_{i,j,k,l,m} \;=\; h_i \cdot \omega_{i,j,k,l,m}, \tag{1}$$

$$\mathbf{D}: \theta''_{i,j,k,l,m} \;=\; \theta'_{i,j,k,l,m} \Big/ \sqrt{\sum_{i,k,l,m} {\theta'_{i,j,k,l,m}}^2}, \tag{2}$$

where $\theta$ and $\theta''$ are the original and modulated weight, respectively. $h_i$ is the *style* corresponding to the $i$th input channel, $j$ is the output channel dimension, and $k, l, m$ denote the spatial dimension of the 3D convolutional filter.

Once we obtain the final feature volume $\mathbf{F}'_{\text{geo}}$, the feature vector $\mathbf{f}'_{\text{geo}} \in \mathbb{R}^{64}$ of each vertex $\mathbf{v}$ in the tetrahedral grid can be obtained through trilinear interpolation. We additionally feed the coordinates of the point $\mathbf{p}$ to a $[\sin(\mathbf{p}), \cos(\mathbf{p})]$ positional encoding (PE) and concatenate the output with the feature vector $\mathbf{f}'_{\text{geo}}$. To decode the concatenated feature vector into the vertex offset $\Delta\mathbf{v} \in \mathbb{R}^3$ or the SDF value $s \in \mathbb{R}$, we pass it through three conditional FC layers (*ModFC* in Figure B). The modulation and demodulation in these layers is done analogously to Eq. 2. All the layers, except for the last, are followed by the leaky-ReLU activation function. In the last layer, we apply `tanh` to either normalize the SDF prediction $s$ to be within $[-1, 1]$, or normalize the $\Delta\mathbf{v}$ to be within $[-\frac{1}{\text{tet-res}}, \frac{1}{\text{tet-res}}]$, where tet-res denotes the resolution of our tetrahedral grid, which we set to 90 in all the experiments.

Note that for simplicity, we remove all the noise vector from StyleGAN [18, 19] and only have stochasticity in the input $\mathbf{z}$. Furthermore, following practices from DEFTET [15] and DMTET [26], we us two copies of the geometry generator. One generates the vertex offsets $\Delta\mathbf{v}$, while the other outputs the SDF values $s$. The architecture of both is the same, except for the output dimension and activation function of the last layer.

**Volume Subdivision:** In cases where modeling at a high-resolution is required (e.g. motorbike with thin structures in the wheels), we further use volume subdivision following DMTET [26]. As illustrated in Fig. A, we first subdivide the tetrahedral grid and compute SDF values of the new vertices (midpoints) by averaging the SDF values on the edge. Then we identify tetrahedra that have vertices with different SDF signs. These are the tetrahedra that intersect with the underlying surface encode by SDF. To refine the surface at increased grid resolution after subdivision, we further predict the residual on SDF values and deformations to update $s$ and $\Delta\mathbf{v}$ of the vertices in **identified** tetrahedra. Specifically, we use an additional 3D convolutional layer to upsample feature volume

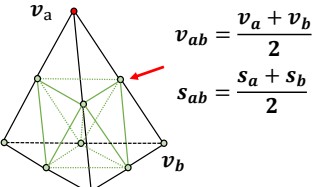

Figure A: With volume subdivision, each tetrahedron is divided into 8 smaller tetrahedra by connecting midpoints.

$$v_{ab} = \frac{v_a + v_b}{2}$$

$$s_{ab} = \frac{s_a + s_b}{2}$$

$\mathbf{F}'_{\text{geo}}$ to $\mathbf{F}''_{\text{geo}}$ of shape $64 \times 64 \times 64 \times 8$ conditioned on $w_1$. Then, following the steps described above, we use trilinear interpolation to obtain per-vertex feature, concatenate it with PE and decode the residuals $\delta s$ and $\delta\mathbf{v}$ using conditional FC layers. The final SDF and vertex offset are computed as:

$$s' = s + \delta s, \; \Delta\mathbf{v}' = \Delta\mathbf{v} + \delta\mathbf{v}. \tag{3}$$

### A.3 Texture Generator

We adapt the generator architecture from StyleGAN2 [19] to generate a tri-plane representation of the texture field. Similar as in the geometry generator, we start from a randomly initialized feature grid $\mathbf{F}_{\text{tex}} \in \mathbb{R}^{4 \times 4 \times 512}$ that is shared across the shapes, and is learned during training. This initial feature grid is up-sampled to a feature grid $\mathbf{F}'_{\text{tex}} \in \mathbb{R}^{256 \times 256 \times 96}$ that is conditioned on $\mathbf{w}_1$ and $\mathbf{w}_2$. Specifically, we use a series of six modulated 2D convolution blocks (*ModBlock2D* in Figure B). The *ModBlock2D* blocks are the same as the *ModBlock3D* blocks, except that the convolution is 2D and that the conditioning is on $\mathbf{w}_1 \oplus \mathbf{w}_2$, where $\oplus$ denotes concatenation. Additionally, the output of each *ModBlock2D* block is passed through a conditional *tTPF* layer that applies a conditional 2D convolution with kernel size 1x1. Note that, following the practices from StyleGAN2 [19], the conditioning in the *tTPF* layers is performed only through *modulation* of the weights (no *demodulation*).

The output of the last *tTPF* layer is then reshaped into three axis-aligned feature planes of size $256 \times 256 \times 32$.

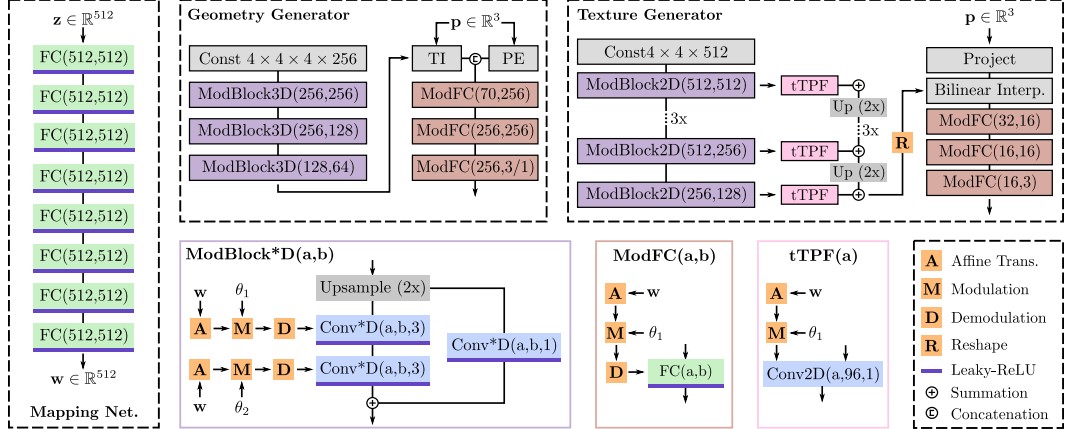

Figure B: **Network architecture of GET3D**. TI and PE denote trilinear interpolation and positional encoding, respectively. FC($a, b$) represents a fully connected layer with $a$ and $b$ denoting the input and output dimension, respectively. Similarly, Conv3D($a, b, c$) denotes a 3D convolutional layer with $a$ input channels, $b$ output channels, and kernel dimension $c \times c \times c$. In the Texture Generator, the block ModBlock2D(512,512) is repeated four times. All convolutional layers have stride 1.

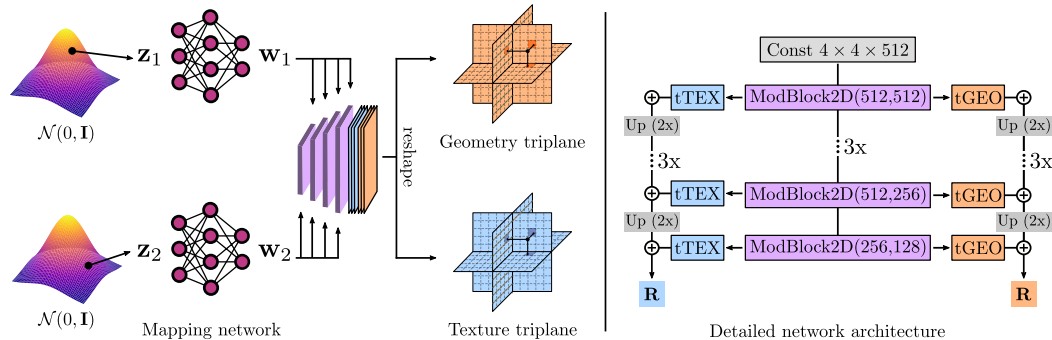

Figure C: **Improved generator architecture of GET3D**. High-level overview of our 3D **Generator** *left* and detailed architecture *right*. Different to the model architecture proposed in the main paper, the new generator shares the same backbone network for both geometry and texture generation. This improves the information flow and enables better disentanglement of the geometry and texture.

To obtain the feature $\mathbf{f}_{\text{tex}} \in \mathbb{R}^{32}$ of a surface point $\mathbf{p} \in \mathbb{R}^3$, we first project $\mathbf{p}$ onto each plane, perform bilinear interpolation of the features, and finally sum the interpolated features:

$$\mathbf{f}_{\text{tex}} = \sum_e \rho(\pi_e(\mathbf{p})), \tag{4}$$

where $\pi_e(\mathbf{p})$ is the projection of the point $\mathbf{p}$ to the feature plane $e$ and $\rho(\cdot)$ denotes bilinear interpolation of the features. Color $\mathbf{c} \in \mathbb{R}^3$ of the point $\mathbf{p}$ is then decoded from $\mathbf{f}^t$ using three conditional FC layers (*ModFC*) conditioned on $\mathbf{w}_1 \oplus \mathbf{w}_2$. The hidden dimension of each layer is 16. Following StyleGAN2 [19], we do not apply normalization to the final output.

## A.4   2D Discriminator

We use two discriminators to train GET3D: one for the RGB output and one for the 2D silhouettes. For both, we use exactly the same architecture as the discriminator in StyleGAN [18]. Empirically, we have observed that conditioning the discriminator on the camera pose leads to canonicalization of the shape orientations. However, discarding this conditioning only slightly affects the performance, as shown in Section C.3. In fact, we primarily use this conditioning to enable the evaluation of geometry using evaluation metrics, which assume that the shapes are generated in the canonical frame.

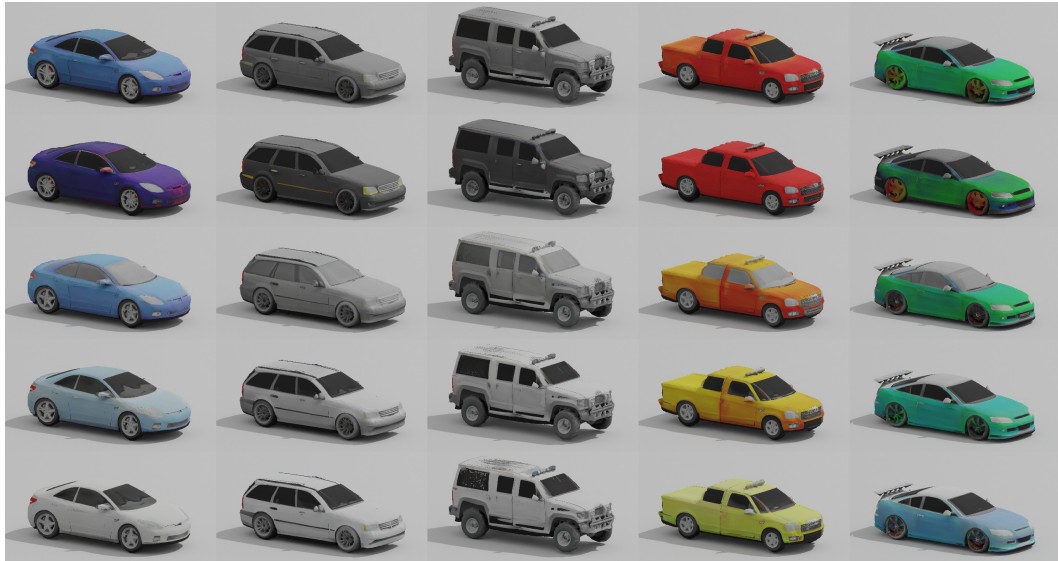

Figure D: **Disentanglement of geometry and texture achieved by the original model depicted in Fig. ??**. In each row, we show shapes generated from the same texture latent code, while changing the geometry latent code. In each column, we show shapes generated from the same geometry latent code, while changing the texture code. The original model fails to achieve good disentanglement.

## A.5   Improved Generator

The motivation for sampling two noise vectors ($\mathbf{z}_1$, $\mathbf{z}_2$) in the generator is to enable disentanglement of the geometry and texture, where geometry is to be treated as a first-class citizen. Indeed, the geometry should only be controlled by the geometry latent code, while the texture should be able to not only adapt to the changes in the texture latent code, but also to the changes in geometry, i.e. a change in the geometry latent should propagate to the texture. However, in the original design of the GET3D generator the information flow from the geometry to the texture generator is very limited—concatenation of the two latent codes (Fig. B). Such a weak connection makes it hard to learn the disentanglement of geometry and texture and the texture generator can even learn to ignore the texture latent code (Fig. D.).

This empirical observation motivated us to improve the design of the generator network, after the initial submission, by improving the information flow, which in turn better supports the disentanglement of the geometry and texture. To this end, our improved generator shares the same backbone network for both geometry and texture generation, as shown in Fig. C. In particular, we follow SemanticGAN [21] and use StyleGAN2 [19] backbone. Each ModBlock2D (modulated with the geometry latent code $\mathbf{w}_1$), now has two tTPF branches, one for generating the geometry feature (*tGEO*), and the other for generating texture features (*tTEX*). The output of this backbone network are two feature triplanes, one for geometry and one for texture. To predict the SDF value and deformation for each vertex in the tetrahedral grid, we project the vertex onto each of the geometry triplanes, obtain its feature vector using Eq. 4, and finally use a ModFC to decode $s_i$ and $\Delta\mathbf{v}_i$. The prediction of the color in the texture branch remains unchanged.

Qualitative result of the geometry and texture disentanglement achieved with this improved generator is depicted in Fig. E and F. Shared backbone network allows us to achieve much better disentanglement of geometry and texture (Fig. D vs Fig. E), while also achieving better quantitative metrics on the task of unconditional generation

## A.6   Training Procedure and Hyperparameters

We implement GET3D on top of the official PyTorch implementation of StyleGAN2 [19][1]. Our training configuration largely follows StyleGAN2 [19] including: using a minibatch standard devia-

---

[1]StyleGan3: `https://github.com/NVlabs/stylegan3` (NVIDIA Source Code License)

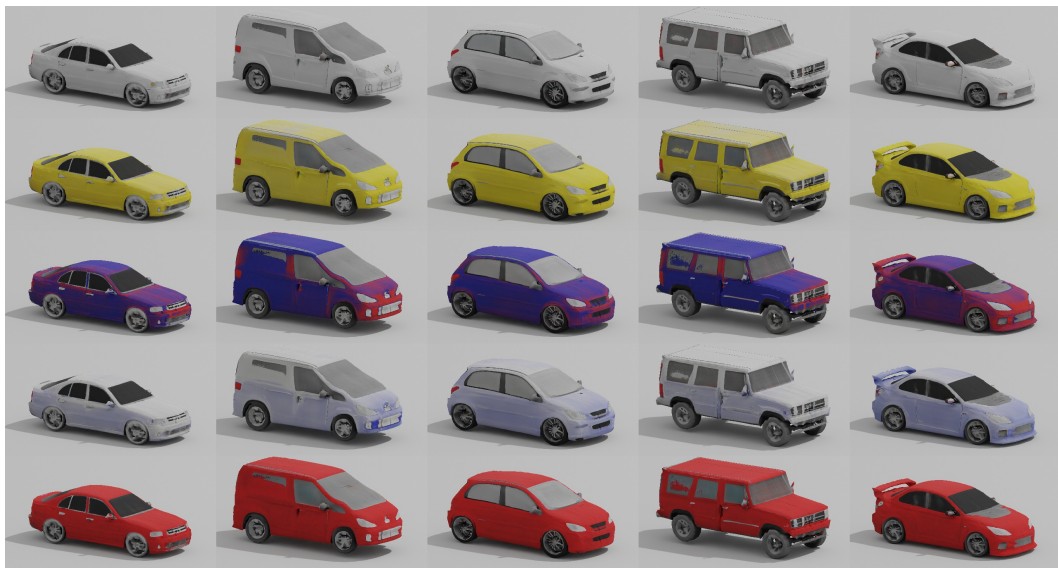

Figure E: **Disentanglement of geometry and texture achieved by the improved model depicted in Fig. C**. In each row, we show shapes generated from the same texture latent code, while changing the geometry latent code. In each column, we show shapes generated from the same geometry latent code, while changing the texture code. The disentanglement in this model is poor. Comparing with Fig. D, this improved model achieves significant better disentanglement of geometry and texture.

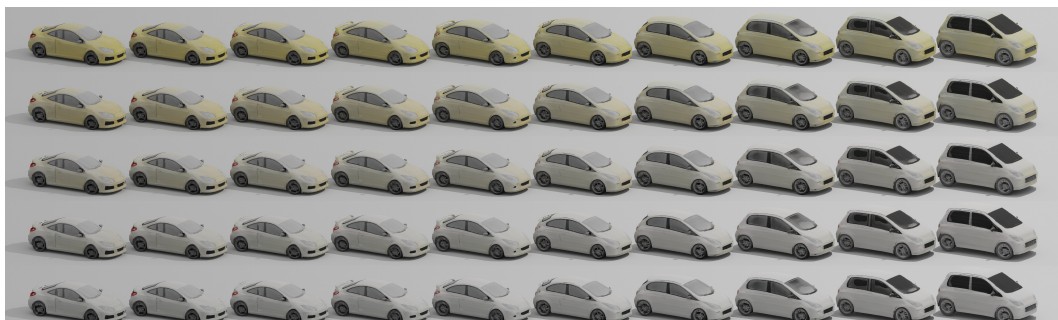

Figure F: **Shape Interpolation.** We interpolate the latent code from top-left corner to the bottom-right corner. In each row, we keep the texture latent code fixed and interpolate the geometry latent code. In each column, we keep the geometry latent code fixed and interpolate the texture latent code. GET3D adequately disentangles geometry and texture, while also providing a meaningful interpolation for both geometry or texture.

tion in the discriminator, exponential moving average for the generator, non-saturating logistic loss, and R1 Regularization. We train GET3D along with the 2D discriminators from scratch, without progressive training or initialization from pretrained checkpoints. Most of our hyper-parameters are adopted form styleGAN2 [19]. Specifically, we use Adam optimizer with learning rate 0.002 and $\beta = 0.9$. For R1 regularization, we set the regularization weight $\gamma$ to 3200 for chair, 80 for car, 40 for animal, 80 for motorbike, 80 for renderpeople, and 200 for house. We follow StyleGAN2 [19] and use lazy regularization, which applies R1 regularization to discriminators only every 16 training steps. Finally, we set the hyperparameter $\mu$ that controls the SDF regularization to 0.01 in all the experiments. We train our model using a batch size of 32 on 8 A100 GPUs for all the experiments. Training a single model takes about 2 days to converge.

# B Experimental Details

## B.1 Datasets

We evaluate GET3D on ShapeNet [7], TurboSquid [2], and RenderPeople [1] datasets. In the following, we provide their detailed description and the preprocessing steps that were used in our evaluation. Detailed statistic of the datasets is available in Table A.

**ShapeNet**[2] [7] contains more than 51k shapes from 55 different categories and is the most commonly used dataset for benchmarking 3D generative models[3]. Prior work [31, 36] typically uses the categories *Airplane*, *Car*, and *Chair* for evaluation. Herein, we replace the category *Airplane* with *Motorcycle*, which has more complex geometry and contains shapes with varying genus. *Car*, *Chair*, and *Motorcycle* contain 7497, 6778, and 337 shapes, respectively. We random split the shapes of each category into training (70%), validation (10%), and test (20%) and remove from the test set shapes that have duplicates in the training set.

**TurboSquid**[4] [2] is a large collection of various 3D shapes with high-quality geometry and texture, and is thus well suited to evaluate the capacity of GET3D to generate shapes with high-quality details. To this end, we use the category *Animal* that contains 442 textured shapes with high diversity ranging from cats, dogs, and lions, to bears and deer [26, 32]. We again randomly split the shapes into training (70%), validation (10%), and test (20%) set. Additionally, we provide qualitative results on the category *House* that contains 563 shapes. Since we perform only qualitative evaluation on *House*, we use all the shapes for training.

**RenderPeople**[5] [1] is a large dataset containing photorealistic 3D models of real-world *humans*. We use it to showcase the capacity of GET3D to generate high-quality and diverse characters that can be used to populate virtual environments, such as games or even movies. In particular, we use 500 models from the whole dataset for training and only perform qualitative analysis.

**Preprocessing** To generate the data, we first scale each shape such that the longest edge of its bounding-box equals $\mathbf{e}_m$, where $\mathbf{e}_m = 0.9$ for *Car*, *Motorcycle*, and *Human*, $\mathbf{e}_m = 0.8$ for *House*, and $\mathbf{e}_m = 0.7$ for *Chair* and *Animal*. For methods that use 2D supervision (Pi-GAN, GRAF, EG3D, and our model GET3D), we then render the RGB images and silhouettes from camera poses sampled from the upper hemisphere of each object. Specifically, we sample 24 camera poses for *Car* and *Chair*, and 100 poses for *Motorcycle*, *Animal*, *House*, and *Human*. The rotation and elevation angles of the camera poses are sampled uniformly from a specified range (see Table A). For all camera poses, we use a fixed radius of 1.2 and the fov angle of $49.13°$. We render the images in Blender [12] using a fixed lighting, unless specified differently.

For the methods that rely on 3D supervision, we follow their preprocessing pipelines [31, 23]. Specifically, for Pointflow [31] we randomly sample 15k points from the surface of each shape, while for OccNet [23] we convert the shapes into watertight meshes by rendering depth frames from random camera poses and performing TSDF fusion.

## B.2 Baselines

**PointFlow** [31] is a 3D point cloud generative model based on continuous normalizing flows. It models the generative process by learning a distribution of distributions. Where the former, denotes the distribution of shapes, and the latter the distribution of points given a shape [31]. PointFlow generates only the geometry, which is represented in the form of a point cloud. To generate the results of [31], we use the original source code provided by the authors[6] and train the models on our data. To compute the metrics based on LFD, we convert the output point clouds (10k points) to a mesh representation using Open3D [37] implementation of Poisson surface reconstruction [20].

---

[2]The ShapeNet license is explained at https://shapenet.org/terms
[3]Herein, we used ShapeNet v1 Core subset obtained from https://shapenet.org/
[4]https://www.turbosquid.com, we obtain consent via an agreement with TurboSquid, and following license at https://blog.turbosquid.com/turbosquid-3d-model-license/
[5]We follow the license of Renderpeople https://renderpeople.com/general-terms-and-conditions/
[6]PointFlow: https://github.com/stevenygd/PointFlow (MIT License)

| Dataset | # Shapes | # Views per shape | Rotation Angle | Elevation Angle |
|---------|----------|-------------------|----------------|-----------------|
| ShapeNet Car | 7497 | 24 | $[0, 2\pi]$ | $[\frac{1}{3}\pi, \frac{1}{2}\pi]$ |
| ShapeNet Chair | 6778 | 24 | $[0, 2\pi]$ | $[\frac{1}{3}\pi, \frac{1}{2}\pi]$ |
| ShapeNet Motorbike | 337 | 100 | $[0, 2\pi]$ | $[\frac{1}{3}\pi, \frac{1}{2}\pi]$ |
| Turbosquid Animal | 442 | 100 | $[0, 2\pi]$ | $[\frac{1}{4}\pi, \frac{1}{2}\pi]$ |
| Turbosquid House | 563 | 100 | $[0, 2\pi]$ | $[\frac{1}{3}\pi, \frac{1}{2}\pi]$ |
| Renderpeople | 500 | 100 | $[0, 2\pi]$ | $[\frac{1}{3}\pi, \frac{1}{2}\pi]$ |

Table A: **Dataset statistics**.

**OccNet** [23] is an implicit method for 3D surface reconstruction, which can also be applied to unconditional generation of 3D shapes. OccNet is an autoencoder that learns a continuous mapping from 3D coordinates to occupancy values, from which an explicit mesh can be extracted using marching cubes [22]. When applied to unconditional 3D shape generation, OccNet is trained as a variational autoencoder. To generate the results of [23], we use the original source code provided by the authors[7] and train the models on our data.

**GRAF** [25] is a generative model that tackles the problem of 3D-aware image synthesis. GRAF's underlying representation is a neural radiance field—conditioned on the shape and appearance latent codes—parameterized using a multi-layer perceptron with positional encoding. To synthesize novel views, GRAF utilizes a neural volume rendering approach similar to Nerf [24]. In our evaluation, we use the source code provided by the authors[8] and train GRAF models on our data.

**Pi-GAN** [5] similar to GRAF, Pi-GAN also tackles the problem of 3D-aware image synthesis, but uses a Siren [27] network—conditioned on a randomly sampled noise vector—to parameterize the neural radiance field. To generate the results of Pi-GAN [5], we use the original source code provided by the authors[9] and train the models on our data.

**EG3D** [6] is a recent model for 3D-aware image synthesis. Similar to our method, EG3D builds upon the StyleGAN formulation and uses a tri-plane representation to parameterize the underlying neural radiance field. To improve the efficiency and to enable synthesis at higher resolution, EG3D utilizes neural rendering at a lower resolution and then upsamples the output using a 2D CNN. The source code of EG3D was provided to us by the authors. To generate the results, we train and evaluate EG3D on our data.

## B.3 Evaluation Metrics

To evaluate the performance, we compare both the texture and geometry of the *generated* shapes $S_g$ to the *reference* ones $S_r$.

### B.3.1 Evaluating the Geometry

To evaluate the geometry, we use all shapes of the test set as $S_r$, and synthesize five times as many generated shapes, such that $|S_g| = 5|S_r|$, where $|\cdot|$ denotes the cardinality of a set. Following prior work [31, 11], we use Chamfer Distance $d_{\text{CD}}$ and Light Field Distance $d_{\text{LFD}}$ [10] to measure the similarity of the shapes, which is in turn used to compute Coverage (COV) and Minimum Matching Distance (MMD) evaluation metrics.

Let $X \in S_g$ denote a generated shape and $Y \in S_r$ a reference one. To compute $d_{\text{CD}}$, we first randomly sample $N = 2048$ points $X_p \in \mathbb{R}^{N \times 3}$ and $Y_p \in \mathbb{R}^{N \times 3}$ from the surface of the shapes $X$ and $Y$, respectively[10] . The $d_{\text{CD}}$ can then be computed as:

$$d_{\text{CD}}(X_p, Y_p) = \sum_{\mathbf{x} \in X_p} \min_{\mathbf{y} \in Y_p} ||\mathbf{x} - \mathbf{y}||_2^2 + \sum_{\mathbf{y} \in Y_p} \min_{\mathbf{x} \in X_p} ||\mathbf{x} - \mathbf{y}||_2^2. \tag{5}$$

---

[7]OccNet: https://github.com/autonomousvision/occupancy_networks (MIT License)

[8]GRAF: https://github.com/autonomousvision/graf (MIT License)

[9]Pi-GAN: https://github.com/marcoamonteiro/pi-GAN (License not provided)

[10]For PointFlow [31], we directly use $N$ points generated by the model.

While Chamfer distance has been widely used in the field of 3D generative models and reconstruction [8, 15, 26], LFD has received a lot attention in computer graphics [10]. Inspired by human perception, LFD measures the similarity between the 3D shapes based on their appearance from different viewpoints. In particular, LFD renders the shapes $X$ and $Y$ (represented as explicit meshes) from a set of selected viewpoints, encodes the rendered images using Zernike moments and Fourier descriptors, and computes the similarity over these encodings. Formal definition of LFD is available in [10]. In our evaluation, we use the official implementation to compute $d_{\text{LFD}}$[11].

We combine these similarity measures with the evaluation metrics proposed in [3], which are commonly used to evaluate 3D generative models:

- **Coverage (COV)** measures the fraction of shapes in the *reference* set that are matched to at least one of the shapes in the *generated* set. Formally, COV is defined as

$$\text{COV}(S_g, S_r) = \frac{|\{\text{argmin}_{X \in S_r} D(X, Y) \,|\, Y \in S_g\}|}{|S_r|}, \tag{6}$$

  where the distance metric D can be either $d_{\text{CD}}$ or $d_{\text{LFD}}$. Intuitively, COV measures the diversity of the generated shapes and is able to detect mode collapse. However, COV does not measure the quality of individual generated shapes. In fact, it is possible to achieve high COV even when the generated shapes are of very low quality.

- **Minimum Matching Distance (MMD)** complements COV metric, by measuring the quality of the individual generated shapes. Formally, MMD is defined as

$$\text{MMD}(S_g, S_r) = \frac{1}{|S_r|} \sum_{X \in S_r} \min_{Y \in S_g} D(X, Y), \tag{7}$$

  where D can again be either $d_{\text{CD}}$ or $d_{\text{LFD}}$. Intuitively, MMD measures the quality of the generated shapes by comparing their geometry to the closest reference shape.

### B.3.2 Evaluating the Texture and Geometry

To evaluate the quality of the generated textures, we adopt the Fréchet Inception Distance (FID) metric, commonly used to evaluate the synthesis quality of 2D images. In particular, for each category, we render 50k views of the generated shapes (one view per shape) from the camera poses randomly sampled from the predefined camera distribution, and use all the images in the test set. We then encode these images using a pretrained Inception v3 [28] model[12], where we consider the output of the last pooling layer as our final encoding. The FID metric can then be computed as:

$$\text{FID}(S_g, S_r) = ||\boldsymbol{\mu}_g - \boldsymbol{\mu}_r||_2^2 + \text{Tr}[\boldsymbol{\Sigma}_g + \boldsymbol{\Sigma}_r - 2(\boldsymbol{\Sigma}_g \boldsymbol{\Sigma}_r)^{1/2}]||, \tag{8}$$

where Tr denotes the trace operation. $\boldsymbol{\mu}_g$ and $\boldsymbol{\Sigma}_g$ are the mean value and covariance matrix of the generated image encoding, while $\boldsymbol{\mu}_r$ and $\boldsymbol{\Sigma}_r$ are obtained from the encoding of the test images.

As briefly discussed in the main paper, we use two variants of FID, which differ in the way in which the 2D images are rendered. In particular, for FID-Ori, we directly use the neural volume rendering of the 3D-aware image synthesis methods to obtain the 2D images. This metric favours the baselines that were designed to directly generate valid 2D images through neural rendering. Additionally, we propose a new metric, FID-3D, which puts more emphasis on the overall quality of the generated 3D shape. Specifically, for the baselines which do not output a textured mesh, we extract the geometry from their underlying neural field using marching cubes [22]. Then, we find the intersection point of each pixel ray with the generated mesh and use the 3D location of the intersected point to query the RGB value from the network. In this way, the rendered image is a more faithful representation of the underlying 3D shape and takes the quality of both geometry and texture into account. Note that FID-3D and FID-Ori are identical for methods that directly generate textured 3D meshes, as it is the case with GET3D.

---

[11]LFD: https://github.com/Sunwinds/ShapeDescriptor/tree/master/LightField/3DRetrieval_v1.8/3DRetrieval_v1.8 (License not provided)

[12]Inception network checkpoint path: http://download.tensorflow.org/models/image/imagenet/inception-2015-12-05.tgz

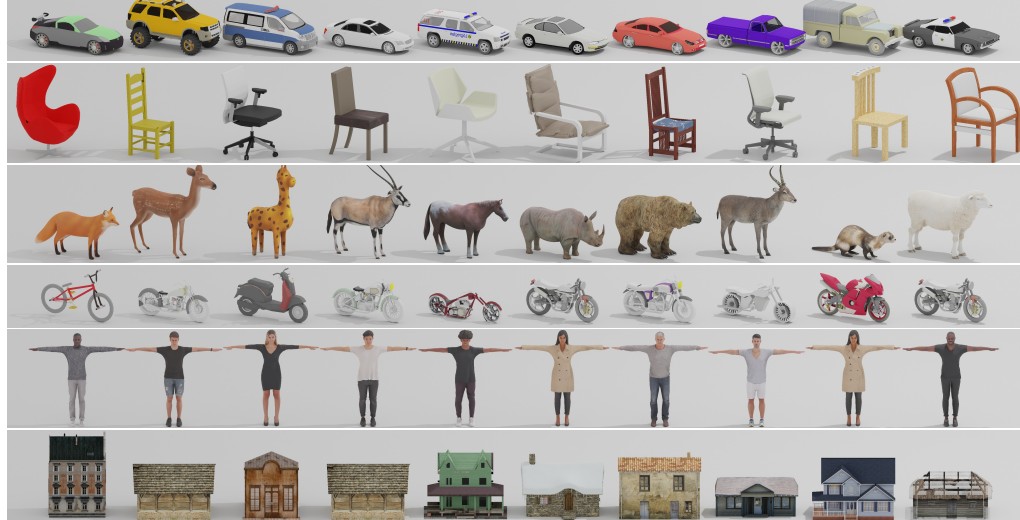

Figure G: **Shape retrieval of our generated shapes**. We retrieve the closest shape in the training set for each of shapes we showed in the Figure 1. Our generator is able to generate novel shapes that are different from the training set

## C Additional Results on the Unconditioned Shape Generation

In this section we provide additional results on the task of unconditional 3D shape generation. First, we perform additional qualitative comparison of GET3D with the baselines in Section C.1. Second, we present further qualitative results of GET3D in Section C.2. Third, we provide additional ablation studies in Section C.3. We also analyse the robustness and effectiveness of GET3D. Specifically, in Sec. C.4 and C.5, we evaluate GET3D trained with noisy cameras and 2D silhouettes predicted by 2D segmentation networks. We further provide addition experiments on StyleGAN generated realistic dataset from GANverse3D [34] in Sec. C.6. Finally, we provide additional comparison with EG3D [6] on human character generation in Sec. C.7.

### C.1 Additional Qualitative Comparison with the Baselines

**Comparing the Geometry of Generated Shapes**    We provide additional visualization of the 3D shapes generated by GET3D and compare them to the baseline methods in Figure Q. GET3D is able to generate shapes with complex geometry, different topology, and varying genus. When compared to the baselines, the shapes generated by GET3D contain more details and are more diverse.

**Comparing the Synthesized Images**    We provide additional results on the task of 2D image generation in Figure R. Even though GET3D is not designed for this task, it produces comparable results to the strong baseline EG3D [6], while significantly outperforming other baselines, such as PiGAN [5] and GRAF [25]. Note that GET3D directly outputs 3D textured meshes, which are compatible with standard graphics engines, while extracting such representation from the baselines is non-trivial.

### C.2 Additional Qualitative Results of GET3D

We provide additional visualizations of the generated geometry and texture in Figures S-X. GET3D can generate high quality shapes with diverse textures across all the categories, from chairs, cars, and animals, to motorbikes, humans, and houses. Accompanying video (*demo.mp4*) contains further visualizations, including detailed 360° turntable animations for 400+ shapes and interpolation results.

**Closest Shape Retrieval**    To demonstrate that GET3D is capable of generating novel shapes, we perform shape retrieval for our generated shapes. In particular, we retrieve the closest shape in the training set for each of shapes we showed in the Figure 1 by measuring the CD between the generated shape and all training shapes. Results are provided in Figure G. All generated shapes in Figure 1

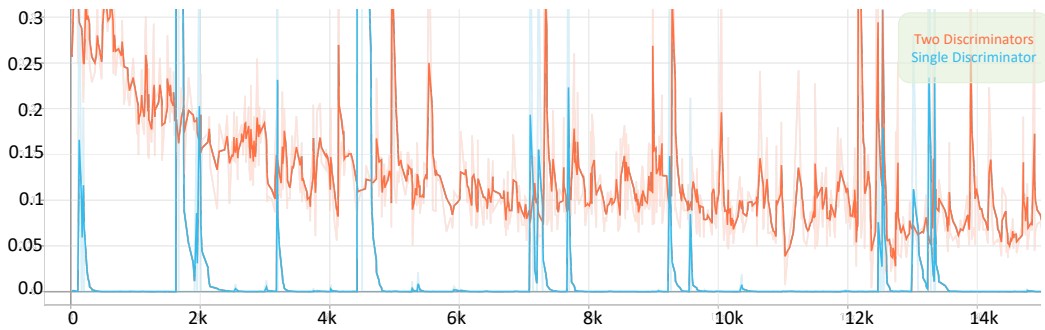

Figure H: **Training loss curve for discriminator.** We compare the training dynamics of using a single discriminator on both RGB image and 2D silhouette, with the ones using two discriminators for each image, respectively. The horizontal axis represents the number of images that the discriminators have seen during training (mod by 1000). Two discriminators greatly reduce training instability and help us obtain good results.

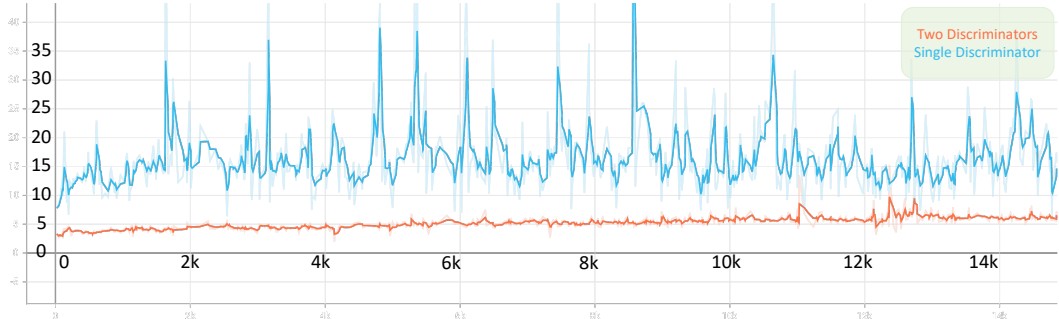

Figure I: **Training loss curve for generator.** We compare the training dynamics for using single discriminator on both RGB image and 2D silhouette with two discriminators for each image, respectively. The horizontal axis represents the number of images discriminator have seen during training (mod by 1000).

significantly differ from their closest shape in the training set, exhibiting different geometry and texture, while still maintaining the quality and diversity.

**Volume Subdivision** We provide further qualitative results highlighting the benefits of volume subdivision in Figure Y. Specifically, we compare the shapes generated with and without volume subdivision on ShapeNet motorbike category. Volume subdivision enables GET3D to generate finer geometric details like handle and steel wire, which are otherwise hard to represent.

### C.3 Additional Ablations Studies

We now provide additional ablation studies in an attempt to further justify our design choices. In particular, we first discuss the design choice of using two dedicated discriminators for RGB images and 2D silhouettes, before ablating the impact of adding the camera pose conditioning to the discriminator.

### C.3.1 Using Two Dedicated Discriminators

We empirically find that using a single discriminator on both RGB image and silhouettes introduces significant training instability, which leads to divergence when training GET3D. We provide a comparison of the training dynamics in Figure H and I, where we depict the loss curves for the generator and discriminator. We hypothesize that the instability might be caused by the fact that a single discriminator has access both geometry (from 2D silhouettes) and texture (from RGB image) of the shape, when classifying whether the image is real or not. Since we randomly initialize our geometry generator, the discriminator can quickly overfit to one aspect—either geometry or

| Model | FID |
|---|---|
| GET3D w.o. Camera Condition | 11.63 |
| GET3D w/ Camera Condition | 10.25 |

Table B: **Ablations on using camera condition**: We ablate using camera condition for discriminator. We train the model on Shapenet Car dataset.

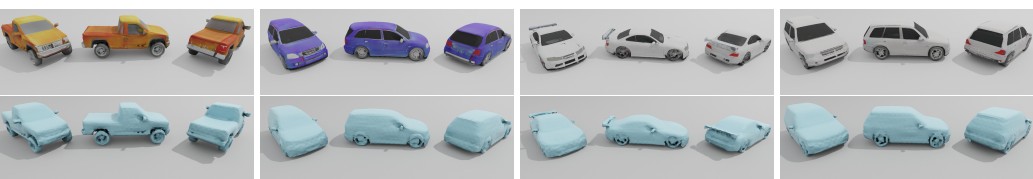

Figure J: **Additional qualitative results of GET3D trained with noisy cameras.** We render generated shapes in Blender. The visual quality is similar to original GET3D in the main paper.

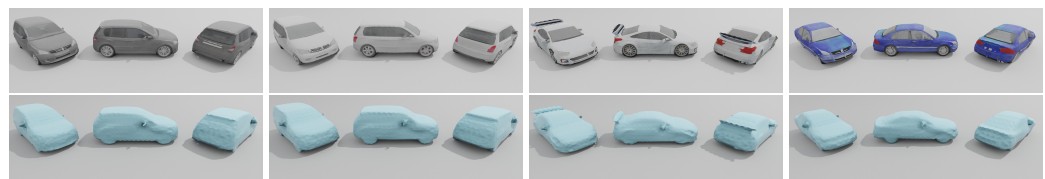

Figure K: **Additional qualitative results of GET3D trained with predicted 2D silhouettes (Mask-Black).** We render generated shapes in Blender. The visual quality is similar to original GET3D in the main paper.

texture—and thus produces bad gradients for the other branch. A two-stage approach in which two discriminators would be used in the first stage of the training, and a single discriminator in the later stage, when the model has already learned to produce meaningful shapes, is an interesting research direction, which we plan to explore in the future.

### C.3.2 Ablation on Using Camera Condition for Discriminator

Since we are mainly operating on synthetic datasets in which the shapes are aligned to a canonical direction, we condition the discriminators on the camera pose of each image. In this way, GET3D learns to generate shapes in the canonical orientation, which simplifies the evaluation when using metrics that assume that the input shapes are canonicalized. We now ablate this design choice. Specifically, we train another model without the conditioning and evaluate its performance in terms of FID score. Quantitative results are given in Table. B. We observe that removing the camera pose conditioning, only slightly degrades the performance of GET3D (-1.38 FID). This confirms that our model can be successfully trained without such conditioning, and that the primary benefit of using it is the easier evaluation.

| Method | FID |
|---|---|
| GET3D - original | 10.25 |
| GET3D - noisy cameras | 19.53 |
| GET3D - predicted 2D silhouettes (Mask-Black) | 29.68 |
| GET3D - predicted 2D silhouettes (Mask-Random) | 33.16 |

Table C: Additional quantitative results for noisy cameras and using predicted 2D silhouettes on Shapenet Car dataset.

### C.4 Robustness to Noisy Cameras

To demonstrate the robustness of GET3D to imperfect cameras poses, we add Gaussian noises to the camera poses during training. Specifically, for the rotation angle, we add a noise sampled from

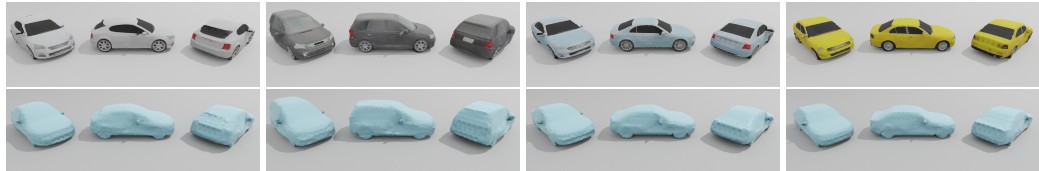

Figure L: **Additional qualitative results of GET3D trained with predicted 2D silhouettes (Mask-Random).** We render generated shapes in Blender. The visual quality is similar to original GET3D in the main paper.

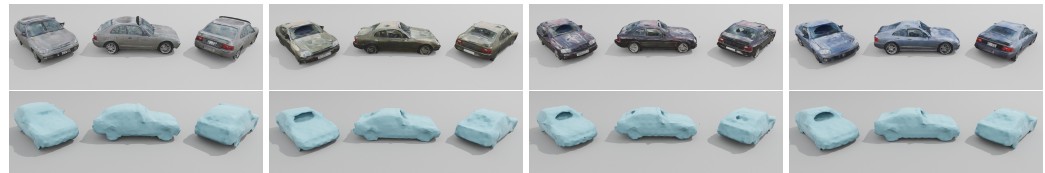

Figure M: **Additional qualitative results of GET3D trained with "real" GANverse3D [34] data.** We render generated shapes in Blender.

a Gaussian distribution with zero mean, and 10 degrees variance. For the elevation angle, we also add a noise sampled from a Gaussian distribution with zero mean, and 2 degrees variance. We use ShapeNet Car dataset [7] in this experiment.

The quantitative results are provided in Table C and qualitative examples are depicted in Figure J. Adding camera noise harms the FID metric, whereas we observe only little degradation in visual quality. We hypothesize that the drop in the FID is a consequence of the camera pose distribution mismatch, which occurs as result of rendering the testing dataset, used to calculate the FID score, with a camera pose distribution without added noise. Nevertheless, based on the visual quality of the generated shapes, we conclude that GET3D is robust to a moderate level of noise in the camera poses.

## C.5 Robustness to Imperfect 2D Silhouettes

To evaluate the robustness of GET3D when trained with imperfect 2D silhouettes, we replace ground truth 2D masks with the ones obtained from Detectron2[13] using pretrained PointRend checkpoint, mimicking how one could obtain the 2D segmentation masks in the real world. Since our training images are rendered with the black background, we use two approaches to obtain the 2D silhouettes: i) we directly feed the original training image into Detectron2 to obtain the predicted segmentation mask (we refer to this as Mask-Black), and ii) we add a background image, randomly sampled from PASCAL-VOC 2012 dataset (we refer to this as Mask-Random). In this setting, the pretrained Detectron2 model achieved 97.4 and 95.8 IoU for the Mask-Black and Mask-Random versions, respectively. We again use the Shapenet Car dataset [7] in this experiment.

**Experimental Results**  Quantitative results are summarized in Table C, with qualitative examples provided in Figures K and L. Although we observe drop in the FID scores, qualitatively the results are still similar to the original results in the main paper. Our model can generate high quality shapes even when trained with the imperfect masks. Note that, in this scenario, the training data for GET3D is different from the testing data that is used to compute the FID score, which could be one of the reasons for worse performance.

## C.6 Experiments on "Real" Image

Since many real-world datasets lack camera poses, we follow GANverse3D [34] and utilize pretrained 2D StyleGAN to generate a realistic car dataset. We train GET3D on this dataset to demonstrate the potential applications to real-world data.

---

[13]https://github.com/facebookresearch/detectron2

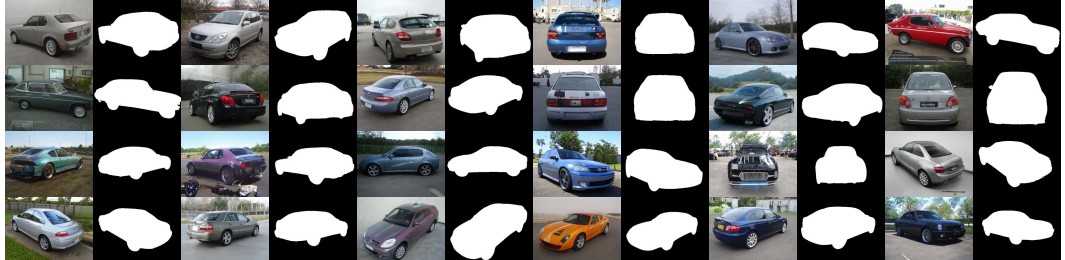

Figure N: We show randomly sampled 2D images and silhouettes from GANverse3D [34] data. Note the realism of the images and the imperfections of the 2D silhouettes.

| Method | FID ($\downarrow$) | |
| --- | --- | --- |
| | Ori | 3D |
| EG3D [6] | **13.77** | 60.42 |
| GET3D | 14.27 | **14.27** |

Table D: Additional quantitative comparison with EG3D [6] on *Human Body* dataset [1].

**Experimental Setting**  Following GANverse3D [34], we manipulate the latent codes of 2D Style-GAN and generate multi-view car images. To obtain the 2D segmentation of each image, we use DatasetGAN [35] to predict the 2D silhouette. We then use SfM [30] to obtain the camera initialization for each generated image. We visualize some examples of this dataset in Fig N and refer the reader to the original GANverse3D paper for more details. Note that, in this dataset both cameras and 2D silhouettes are imperfect.

**Experimental Results**  We provide qualitative examples in Fig. M. Even when faced with the imperfect inputs during training, GET3D is still capable of generating reasonable 3D textured meshes, with variation in geometry and texture.

### C.7  Comparison with EG3D on Human Body

Following the suggestion of the reviewer, we also train EG3D model on the *Human Body* dataset rendered from Renderpeople [1] and compare it to the results of GET3D.

Quantitative results are available in Table D and qualitative comparisons in Figure O. GET3D achieves comparable performance to EG3D [6] in terms of generated 2D images (FID-ori), while significantly outperforming it on 3D shape synthesis (FID-3D). This once more demonstrates the effectiveness of our model in learning actual 3D geometry and texture.

## D  Material Generation for View-dependent Lighting Effects

In modern computer graphics engines such as Blender [12] and Unreal Engine [17], surface properties are represented by material parameters crafted by graphics artists. To make the generated assets graphics-compatible, one direct extension of our method is to also generate surface material properties. In this section, we describe how GET3D is able to incorporate physics-based rendering models, predicting SVBRDF to represent view-dependent lighting effects such as specular surface reflections.

As described in main paper Sec.4.3.1, two modules need to be adapted to facilitate material generation. Namely, the texture generation and the rendering process. Specifically, we repurpose the texture generator branch to predict the Disney BRDF properties [4, 17] on the surface as a reflectance field. Specifically, the texture generator now outputs a 5-channel reflectance property, including surface base color $\mathbf{c}_{base} \in \mathbb{R}^3$, roughness $\beta \in \mathbb{R}$ and metallic $m \in \mathbb{R}$ parameters.

Note that different from a texture field, rendering the reflectance field requires one additional shading step after rasterization into the G-buffer. Thus, the second adaptation is to replace the texture rasterization with an expressive rendering model capable of rendering the reflectance field. According

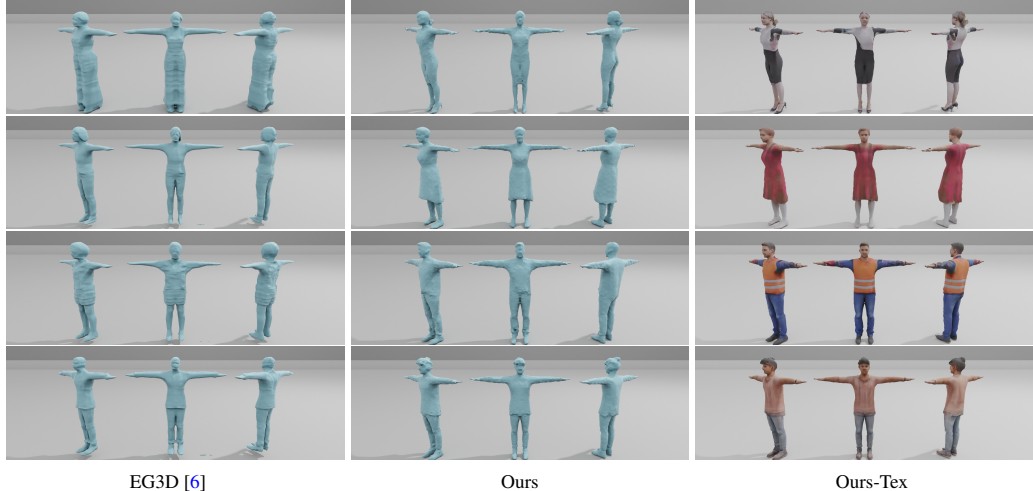

| EG3D [6] | Ours | Ours-Tex |

Figure O: **Additional qualitative comparison on *Human Body* dataset.** We compare our method with EG3D [6] on the extracted geometry.

to the non-emissive rendering equation [16], the outgoing radiance $L_{\mathrm{o}}$ at the camera direction $\boldsymbol{\omega}_{\mathrm{o}}$ is given by:

$$L_{\mathrm{o}}(\boldsymbol{\omega}_{\mathrm{o}}) = \int_{\mathcal{S}^2} L_{\mathrm{i}}(\boldsymbol{\omega}_{\mathrm{i}}) f_{\mathrm{r}}(\boldsymbol{\omega}_{\mathrm{i}}, \boldsymbol{\omega}_{\mathrm{o}}; \mathbf{c}_{\mathrm{base}}, \beta, m)(\mathbf{n} \cdot \boldsymbol{\omega}_{\mathrm{i}})^+ \, \mathrm{d}\boldsymbol{\omega}_{\mathrm{i}}, \tag{9}$$

where $L_{\mathrm{i}}$ is the incoming radiance, $f_{\mathrm{r}}$ is the BRDF, $\mathbf{n}$ is the normal direction on the surface points, $\mathbf{n} \cdot \boldsymbol{\omega}_{\mathrm{i}}$ is the cosine foreshortening term, $\boldsymbol{\omega}_{\mathrm{i}}$ is incoming light direction sampled on sphere $\mathcal{S}^2$, while $(\mathbf{n} \cdot \boldsymbol{\omega}_{\mathrm{i}})^+ = \max(\mathbf{n} \cdot \boldsymbol{\omega}_{\mathrm{i}}, 0)$ constrains the integration over the positive hemisphere. Standard ray tracing technique adopts Monte Carlo sampling methods to estimate this integral, but this incurs large computation and memory cost. Inspired by [29, 33, 9], we instead employ a spherical Gaussian (SG) rendering framework [9], which approximates every term in Eq. (9) with SGs and allows us to analytically compute the outgoing radiance without sampling any rays, from where we can obtain the RGB color for each pixel in the image. We refer the reader to [9] for more details.

Similar to the original training pipeline, we randomly sample light from a set of real-world outdoor HDR panoramas (detailed in the following "Datasets" paragraph) and render the generated 3D assets into 2D images using cameras sampled from the camera distribution of training set. We train the model using the same method as in the main paper by adopting the discriminators to encourage the perceptual realism of the rendered images under arbitrary real-world lighting, along with a second discriminator on the 2D silhouettes to learn the geometry. Note that no supervision from material ground truth is used during training, and the material decomposition emerges in a fully unsupervised manner. When equipped with a physics-based rendering models, GET3D successfully predicts reasonable surface material parameters, generating delicate models which can be directly utilized in stand rendering engines like Blender [12] and Unreal [17].

**Datasets** We collect a set of 724 outdoor HDR panoramas from HDRIHaven[14], DoschDesign[15] and HDRMaps[16], which cover a diverse range of real-world lighting distribution for outdoor scenes. We also apply random flipping and random rotation along azimuth as data augmentation. During training, we convert all the environment maps to SG lighting representations, where we adopt 32 SG lobes, optimizing their directions, sharpness and amplitudes such that the approximated lighting is close to the environment map. We optimize 7000 iterations with MSE loss and Adam optimizer. The converged SG lighting can preserve the most contents in the environment map.

As ShapeNet dataset [7] does not contain consistent material definition, we additionally collect 1494 cars from Turbosquid [2] with materials consistently defined with Disney BRDF. To render the dataset using Blender [12], we follow the camera configuration of ShapeNet Car dataset, and randomly select

[14] polyhaven.com/hdris (License: CC0)

[15] doschdesign.com (License: doschdesign.com/information.php?p=2)

[16] hdrmaps.com (License: Royalty-Free)

from the collected set of HDR panoramas as lighting. In the dataset, the groundtruth roughness for car windows is in the range of $[0.2, 0.4]$ and the metallic is set to $1$; for car paint, the groundtruth roughness is in the range of $[0.3, 0.6]$ and the metallic is set to $0$. We disable complex materials such as the transparency and clear coat effects, such that the rendered results can be interpreted by the basic Disney BRDF properties including base color, metallic and roughness.

**Evaluation metrics**   Since we aim to generate 3D assets that can be used in graphics workflow to produce realistic 2D renderings, we quantitatively evaluate the realism of the 2D rendered images under real-world lighting using FID score.

**Comparisons**   To the best of our knowledge, up to date no generative model can directly generate complex geometry (meshes) with material information. We therefore only compare different version of our model. In particular, we compare the results to the texture prediction version of GET3D, where we do not use material and directly predict RGB color for the surface points. We then ablate the effects of using real-world HDR panoramas for lighting, which are typically hard to obtain. To this end, we manually use two spherical Gaussians for ambient lighting and a random directions to simulate the lighting when rendering the generated shapes during training, and try to learn the materials under this simulated lighting.

**Results**   The quantitative FID scores are provided in Table E. With material generation, the FID score improves by more than 2 points when compared to the texture prediction baseline (18.53 vs 20.78). This indicates that the material generation version of GET3D has better capacity and improved realism compared to the texture only baseline. When using the simulated lighting, instead of real-world HDR panorama, the FID score gets slightly worse but still produces reasonable performance. We further provide additional qualitative results in Fig. P visualizing rendering results of generated assets under different real-world lighting conditions. We import our generated assets in Blender and show animated visualization in the accompanied video (*demo.mp4*).

| Method | FID |
|---|---|
| Ours (Texture) | 20.78 |
| Ours + Material (Ambient and directional light) | 22.83 |
| Ours + Material (Real-world light) | **18.53** |

Table E: Quantitative FID results of material generation.

# E   Text-Guided 3D Synthesis

**Technical details.**   As briefly described in Sec.4.3.2, our text-guided 3D synthesis method follows the dual-Generator design from StyleGAN-NADA [14], and uses the directional CLIP loss [14]. In particular, at each optimization iteration, we randomly sample $N = 16$ camera views and render $N$ paired images using two generators: the frozen one ($G_f$) and the trainable one ($G_t$). The directional CLIP loss can then be computed as:

$$L_{clip} = 1 - \frac{1}{N} \sum_{i=1}^{N} \frac{\Delta I_i \cdot \Delta T}{|\Delta I_i| \cdot |\Delta T|} \tag{10}$$

where $\Delta I_i = E(R(G_t(w), c_i)) - E(R(G_f(w), c_i))$ is the translation of the CLIP embeddings ($E$) from the rendering with $G_f$ to the rendering with $G_t$, under camera $c_i$ and $\Delta T$ is the CLIP embedding translation from the class text label to the provided query text. In our implementation, we used two pre-trained CLIP models with different Vision Transformers ('ViT-32/B' and 'ViT-B/16') [13] for different level of details, and follow the text augmentation as in the StyleGAN-NADA codebase[17].

---

[17]https://github.com/rinongal/StyleGAN-nada (MIT License)

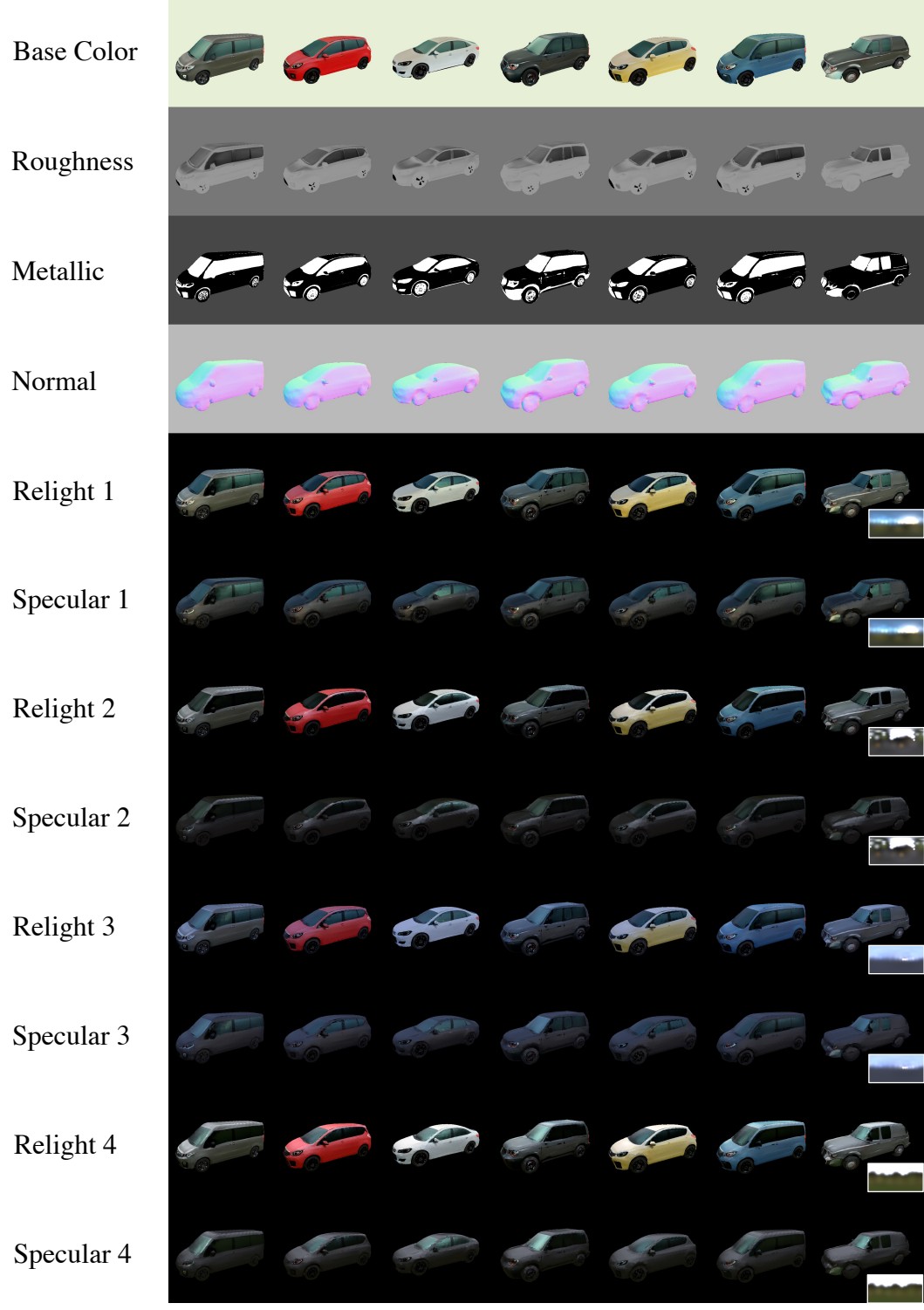

Figure P: **Material generation and relighting.** We visualize seven generated cars' material properties and relight with four different lighting conditions.

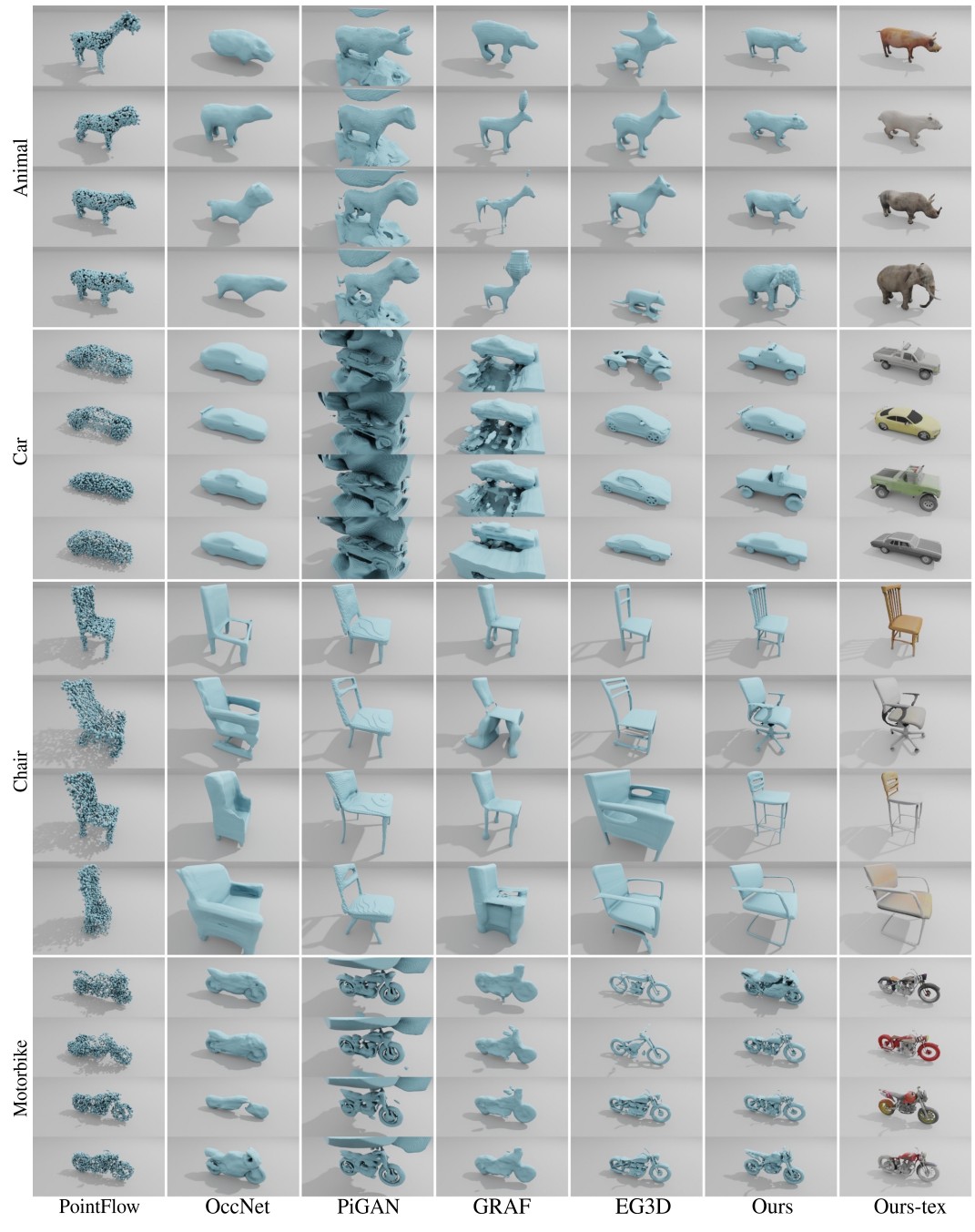

Figure Q: **Generated 3D Geometry.** Additional qualitative comparison with baseline methods on generated 3D geometry

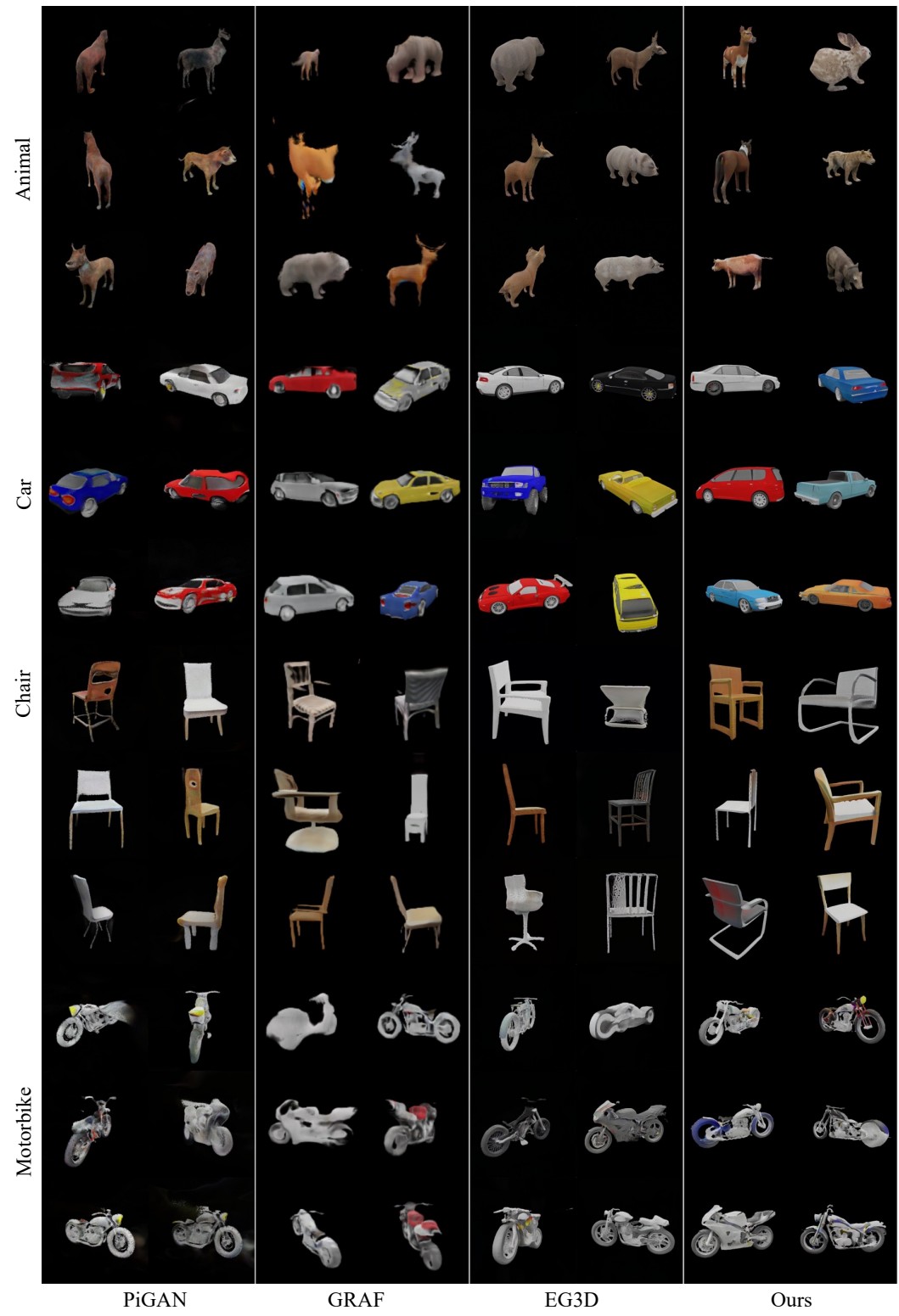

Figure R: **Generated Image.** Additional qualitative comparison with baseline methods on generated 2D images.

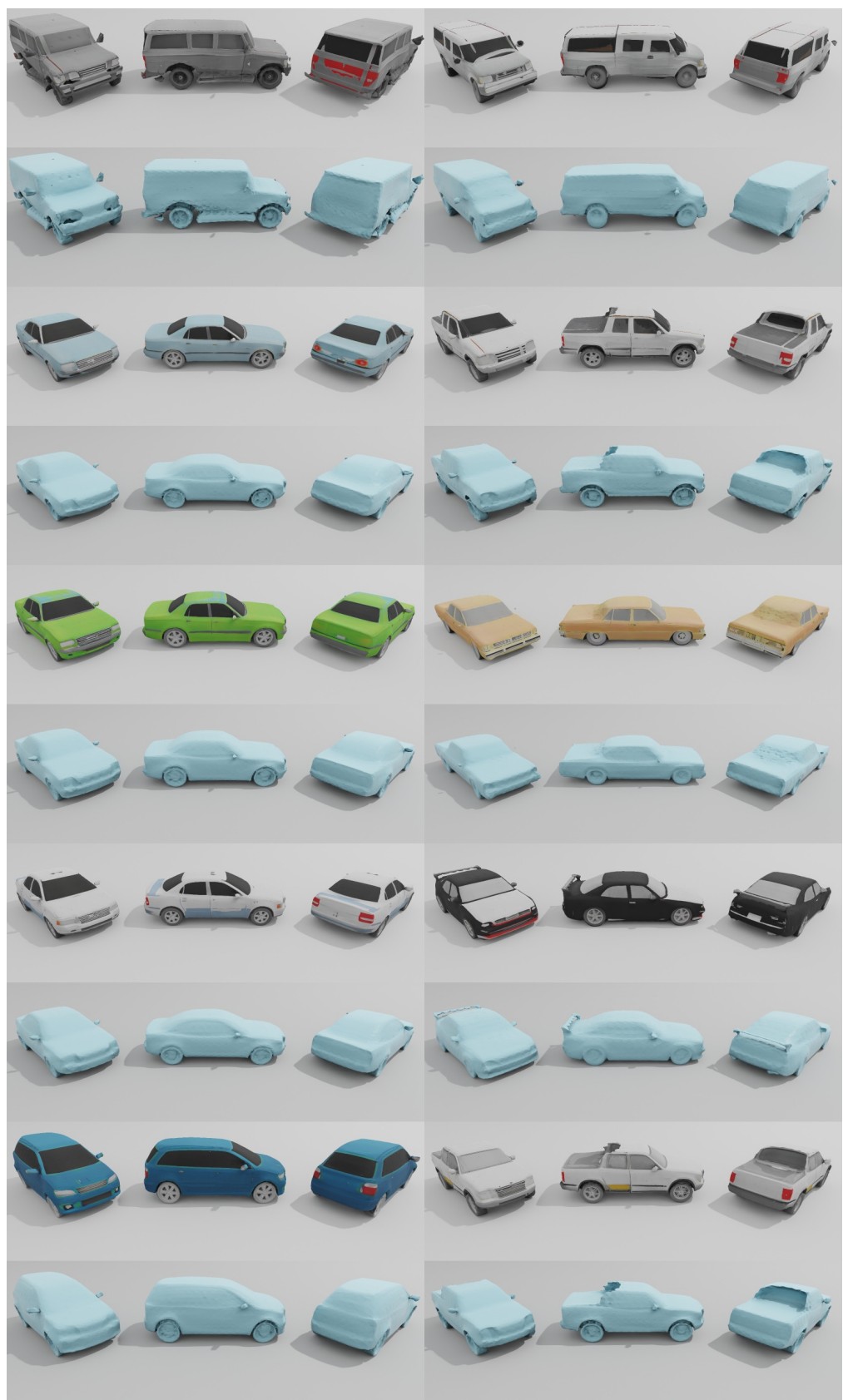

Figure S: Qualitative results on ShapeNet cars.

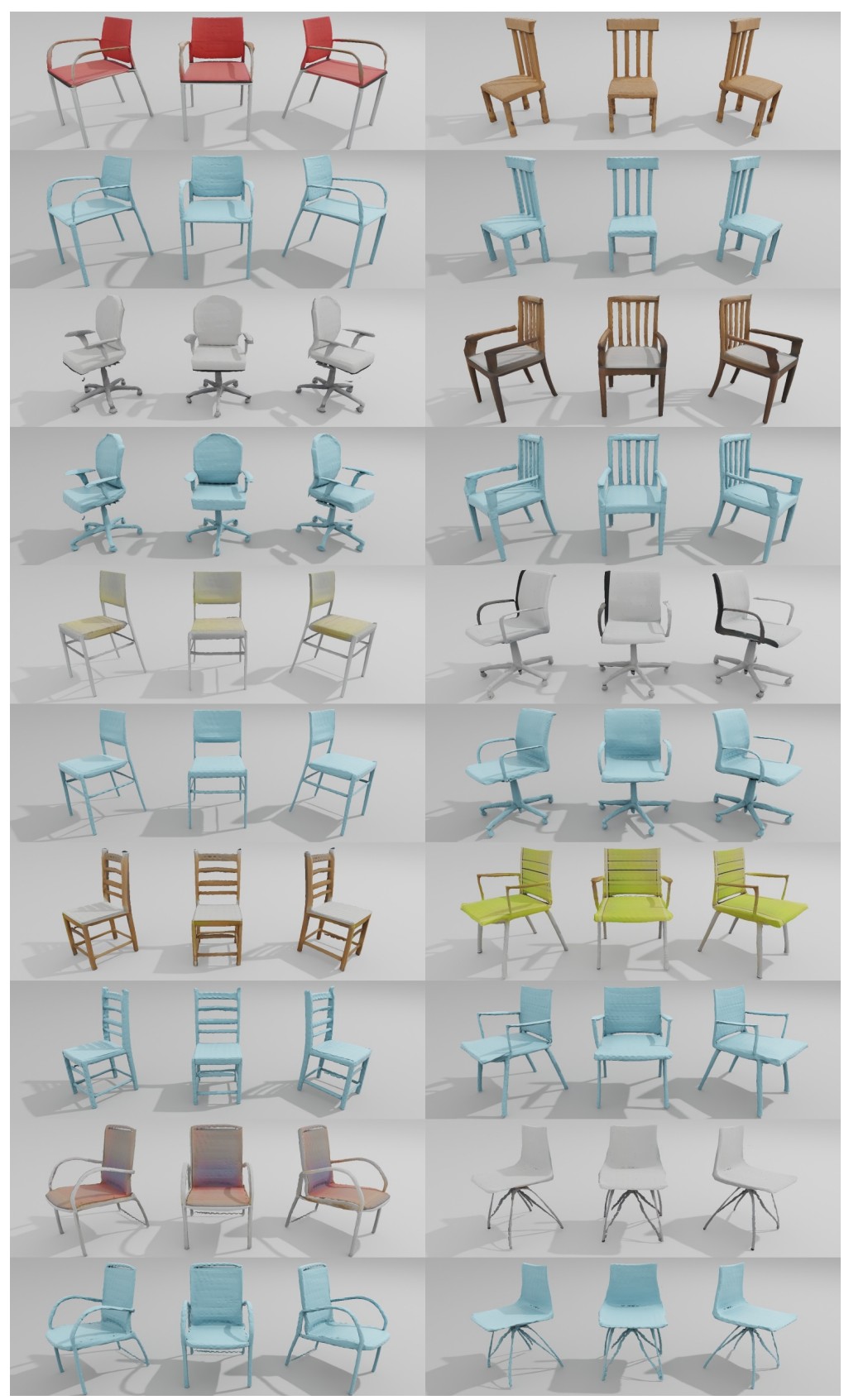

Figure T: Qualitative results on ShapeNet chairs.

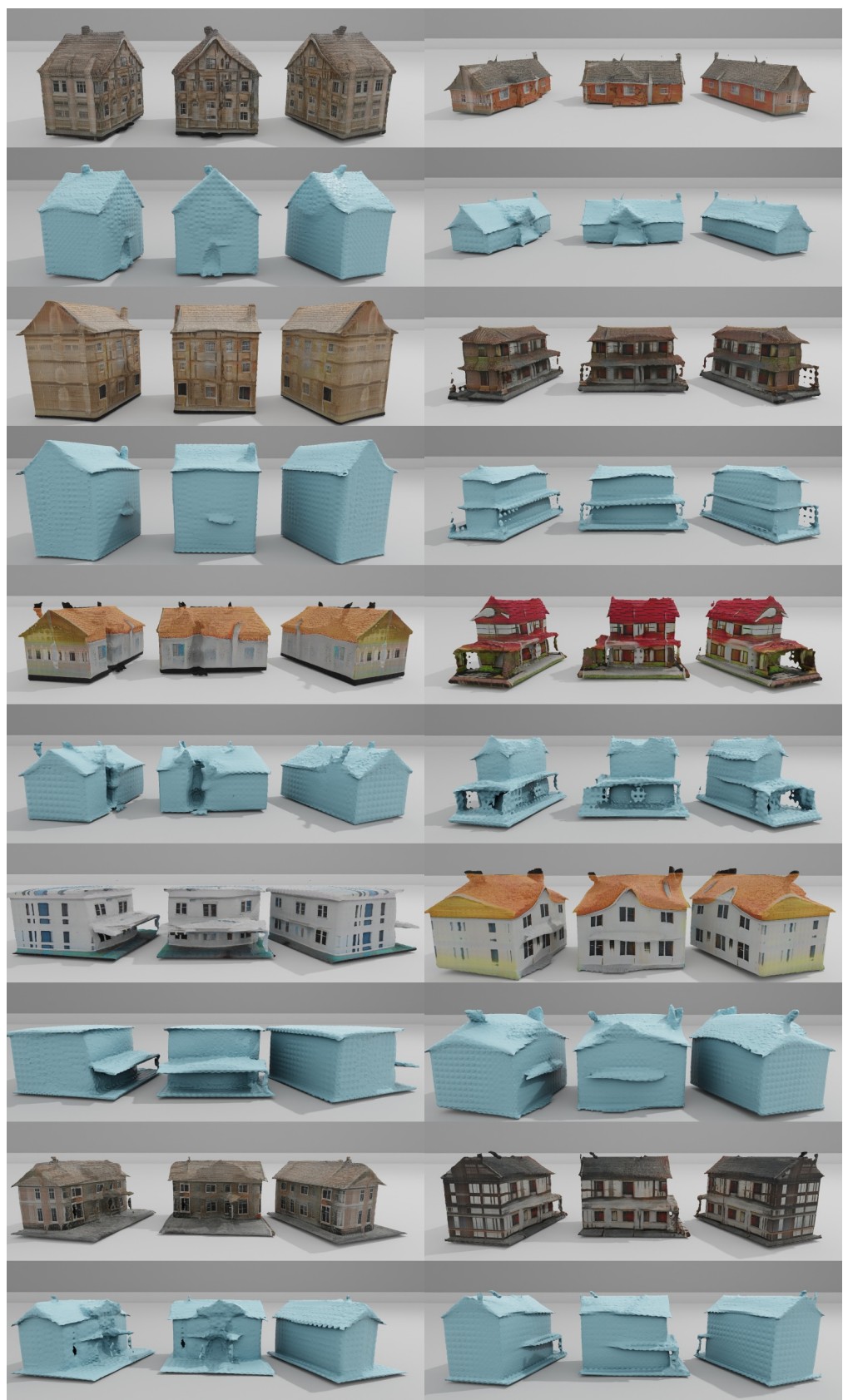

Figure U: Qualitative results on Turbosquid houses.

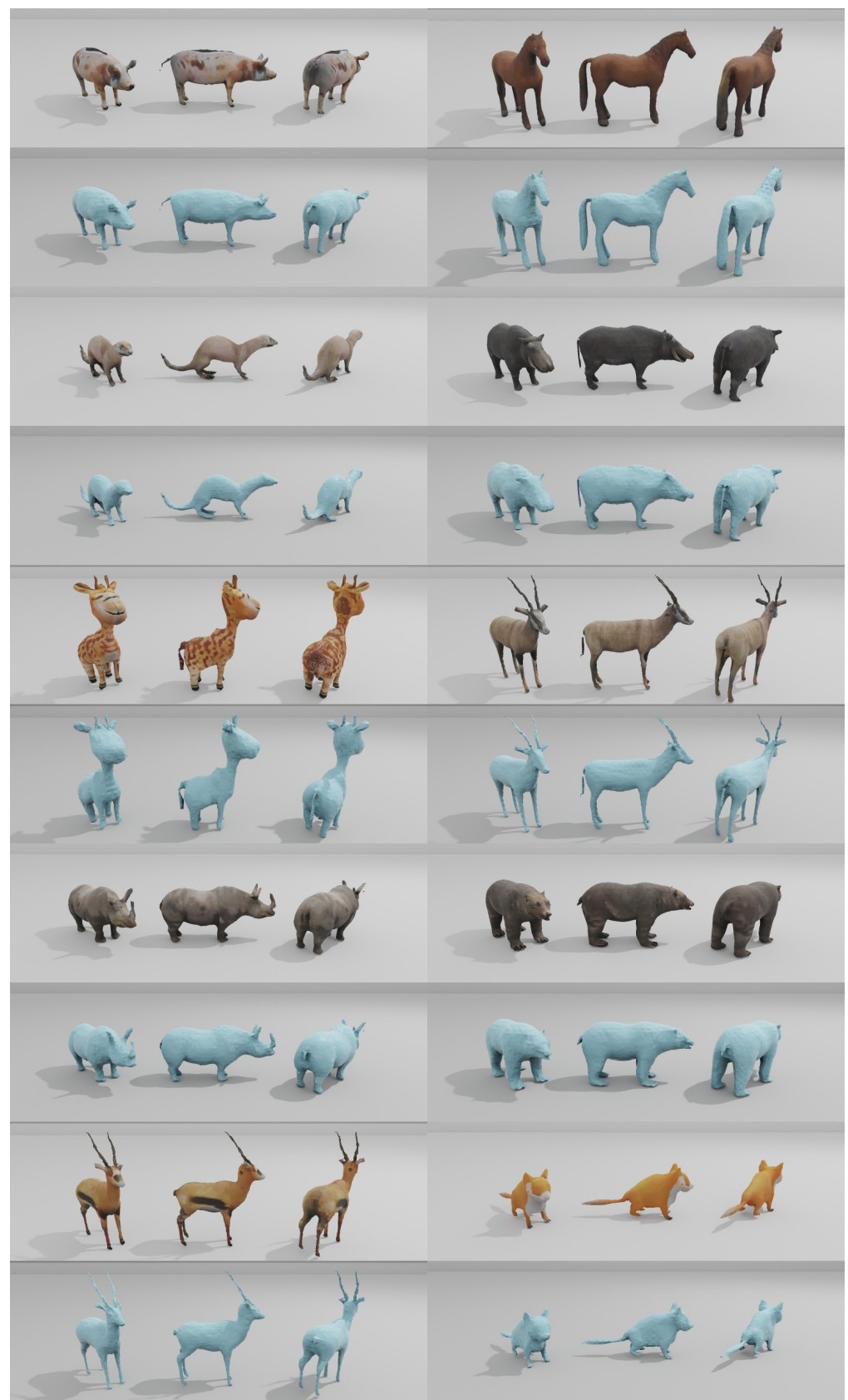

Figure V: Qualitative results on Turbosquid animals.

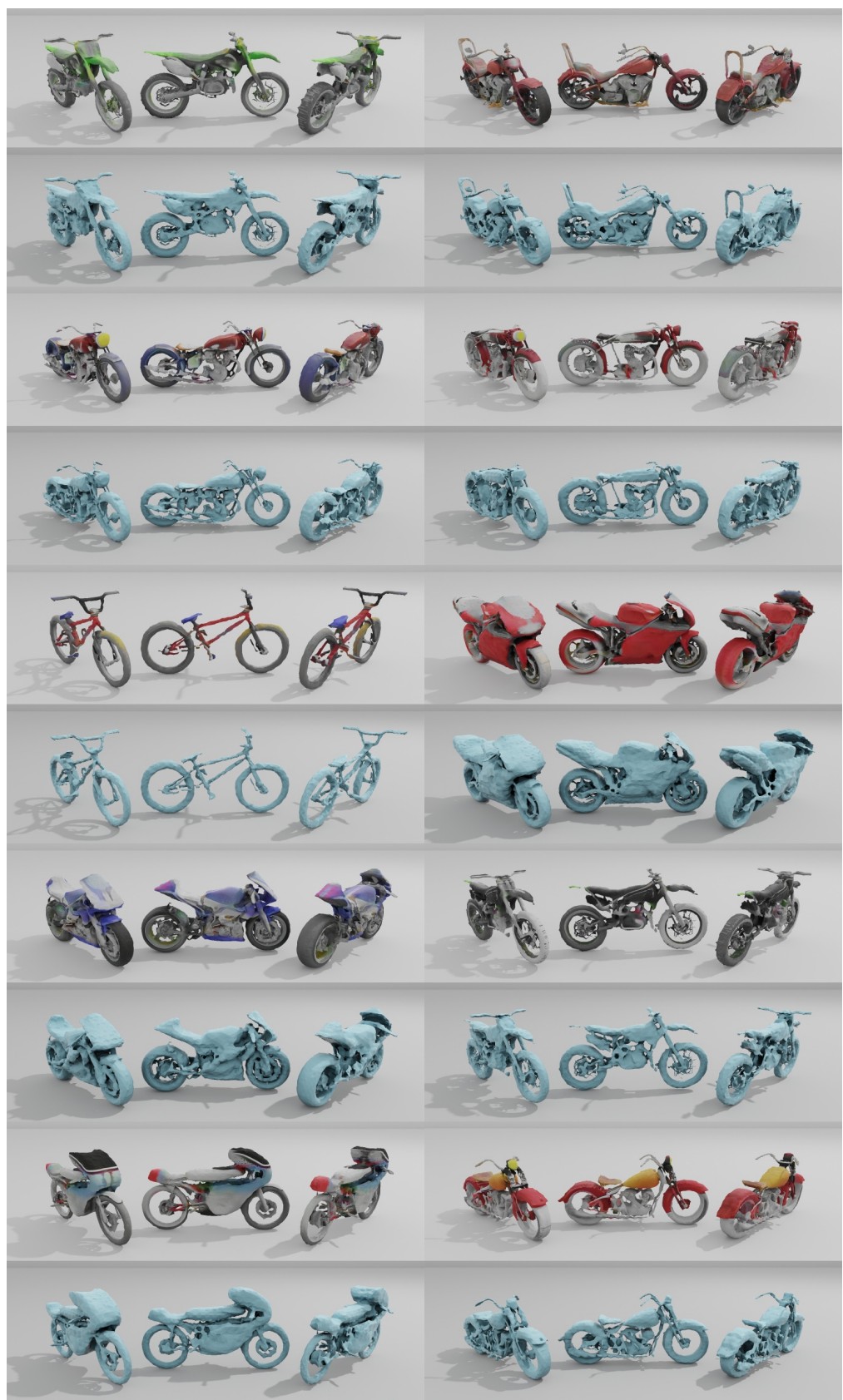

Figure W: Qualitative results on ShapeNet motorbikes.

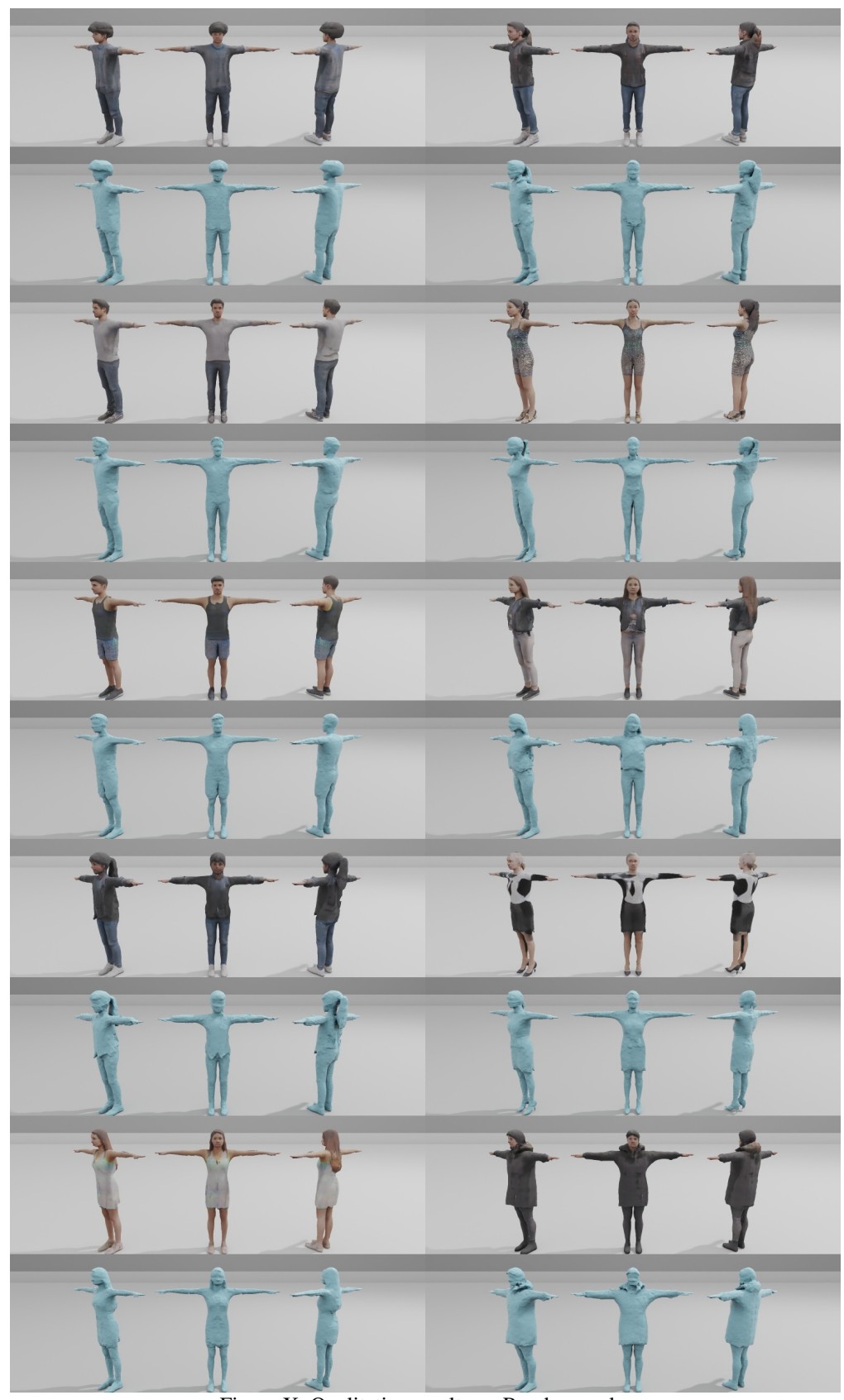

Figure X: Qualitative results on Renderpeople.

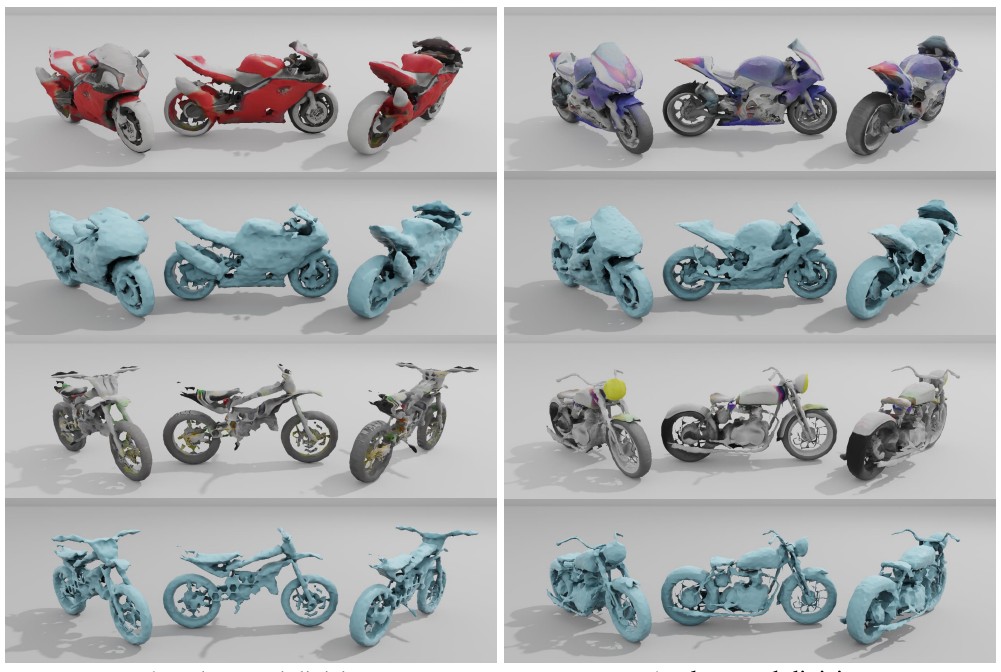

w/o volume subdivision                    w/ volume subdivision

Figure Y: We compare results with and without applying volume subdivision on ShapeNet motorbikes. With volume subdivision, our model can generate finer geometric details like handle and steel wire.