# OpenReview forum: "GET3D: A Generative Model of High Quality 3D Textured Shapes Learned from Images"
_NeurIPS.cc/2022/Conference — NeurIPS 2022 Accept_

### Official Review · Reviewer_LrUW · 2022-06-21

**Rating:** 7
**Confidence:** 3
**Soundness:** 4 excellent
**Presentation:** 4 excellent
**Contribution:** 3 good

**Summary:**

- A generative model of 3D models for 3D content creation should output a mesh of detailed geometry and any topology and must be trained using 2D images without 3D supervision. The proposed method is the first that has these properties. The proposed realizes it by combining a 3D representation of meshes with any topology (DMTet) [51] (S.3.1.1), texture fields [43] (S.3.1.2), differentiable rendering [31], and StyleGAN [29] (S.3.2).
- The experiments conducted on ShapeNet, Turbosquid, and Renderpeople demonstrate that the variety and quality of generated 3D models by the proposed method are better than existing 3D mesh generation models and 3D-aware image generation models. Also, the proposed method is applied to (1) unsupervised decomposition of geometry, light, and material, and (2) text-guided 3D model generation.

**Questions:**

It is written that it is difficult to extract meshed textures from neural fields (l.45, l.85), but what exactly is the difficulty? That is understandable when volume rendering is used, but when SDF is used, I guess the difficulty is the same as the proposed method using texture fields.

**Limitations:**

It is described that it is difficult to apply the proposed method to natural datasets because the distribution of camera poses and silhouettes images are assumed to be known. No other serious limitations were found.

**Strengths And Weaknesses:**

#### Strength

- The proposed method can handle topology changes as it uses DMTet as a 3D representation, different from The existing generative model of meshes [46]. It is a significant step toward realistic 3D content generation models.
- Comparison with existing 3D generation methods [46,11] are conducted on multiple datasets, and quantitative improvement is certain in the geometry diversity and visual quality metrics. Also, it is a bit surprising that the proposed method generates better images than recent 3D-aware image generation models inspired by NeRF.
- Qualitative results are impressive and clearly outperform the baselines. Improvement of texture is explained by the image resolution in training (Table 3), which is also an interesting finding.
- Though no quantitative comparison is provided, the quality of text-guided 3D model generation is significantly better than Text2Mesh [37].

#### Weakness

- The proposed method is a straightforward combination of existing methods [51,43,31,29], and each component is technically not novel, although what is realized by the combination is novel.
- Training the proposed method requires the distribution of camera parameters and silhouette images, so it practically requires a synthetic dataset of 3D models, as noted in l.300. Therefore, the cost of training datasets is still not very lower than existing methods that require 3D supervision.
- The generated shape and texture look a bit coarse when zoomed up, though they are finer than [46,11].
- DMTet is not very good at representing high-resolution scenes as it stores information in 3D grids. Therefore, it is difficult by the proposed method to generate a large-scale scene containing multiple objects, such as bedrooms, unlike 2D GANs.
- The result of unsupervised geometry, lighting, and material decomposition look appealing, but I am not very sure the proposed method is actually good because quantitative comparison is not provided.

Although the technical contribution is not very strong, I recommend acceptance because of the impressive results shown in the experiment.

---

> ### Author Response · Authors · 2022-08-02
> **Thank you for your valuable comments!**
>
> We thank the reviewer for the positive feedback and for perceiving our method as a significant step toward realistic 3D content generation models.  Below, we reply to individual comments and questions raised by the reviewer:
>
> 1. **Novelty:** While we acknowledge that individual submodules of GET3D might not be technically novel on their own,  the novelty of our work lies in devising a proper solution to tackle a practically important research problem. Our paper is motivated by practical problems in 3D content creation and we address many limitations of previous work, by smartly combining the modules, which exploit the recent advances in differentiable surface extraction, highly efficient differentiable rasterization, and GAN-based generative models. As a result, GET3D is not only the first method that achieves such high-quality 3D assets generation, but it is, to the best of our knowledge, also the first  3D generative model that is able to directly output textured meshes with arbitrary topology. We believe that our model opens a new door for 3D generative models and can motivate many follow-up works along this line.
>
> 2. **Supervision cost:** To alleviate the reviewer’s concerns, we have tried to imitate how people would collect the dataset and train GET3D in the real world. To this end, we ran three additional experiments to verify the robustness of our model (See quantitative and qualitative results in Sections 6.1 & 6.2 & 6.3 of the main paper). In particular, we add Gaussian noise to the camera parameters, use a 2D segmentation network to obtain 2D masks and train on StyleGAN generated realistic dataset following GANverse3D [56]. In these challenging settings, GET3D still performs robustly and achieves comparable qualitative results as in our main evaluation. These results also imply that GET3D is a good candidate for 3D textured shape generation from real-world images — an application that we will pursue in our future work. The supervision cost for the training on the StyleGAN dataset is significantly lower than methods that require 3D supervision.
>
> 3. **Room for improvement:** We agree with the reviewer that there is still room for improvement and we wouldn’t dare to assume/imply that GET3D has solved the field of the generative modeling of  3D textured shapes. However, we do believe that our method made a significant step in this direction and that our performance is significantly better than other baselines and creates a new SOTA for generating 3D textured meshes.
>
> 4. **High-resolution scene:** We agree that DMTet itself has the limitation of representing large-scale scenes. However, one potential solution to mitigate this problem is to combine  DMTet with a scene graph representation [a], where we can use the scene graph to represent spatial relations between individual objects and each object can be represented with DMTet. This also opens several interesting opportunities for compositionality, where assets from different datasets could be combined to generate novel scenes. Developing a 3D generative model for scenes is definitely an interesting future direction for our method.
>
> 5. **Quantitative results of material generation:** We provided a quantitative comparison in the Supplementary Material (Section D, Table C), where we evaluate the realism of the 2D rendered images under real-world lighting using the FID scores. Results indicate that the material generation has better capacity and improves realism compared to the texture baseline.
>
> 6. **Difficulty to extract meshed textures from neural fields:** Yes, the difficulty exists when using volume rendering to render the neural fields. In particular, when volume rendering is utilized, it has ambiguities in defining the underlying texture field [b,c] and the texture will not lie exactly on the surface, this is also the reason why people often see foggy results in Nerf [b,c]. Theoretically, when SDF is used, it will not have this issue. However, practically, people often parameterized SDF using neural network, and the generated “SDF” is not guaranteed to be the exact SDF (e.g. one typical approach is utilizing Eikonal regularization on generated SDF [44], but it still can not guarantee the output is a valid SDF), therefore, many papers are still utilizing volume rendering to render the generated “SDF” and they will have the same ambiguity problem as above. On the contrary,  our method does not have this difficulty since we directly render the surface points (L152-L156) and the texture is defined exactly on the object’s surface, despite the fact that we generate a texture field.
>
> If there are any further concerns, we would be happy to address them in further discussion. Thank you!
>
> [a] Neural Scene Graphs for Dynamic Scenes
>
> [b] NeRF++: Analyzing and Improving Neural Radiance Fields
>
> [c] Geo-Neus: Geometry-Consistent Neural Implicit Surfaces Learning for Multi-view Reconstruction.

---

> > ### Comment · Reviewer_LrUW · 2022-08-09
> > **To author feedback**
> >
> > I checked new experimental results. It is good to see that the proposed method works well in a more realistic experimental setting (e.g. natural images, inaccurate supervision).
> >
> > Thank you for telling the qualitative evaluation of the decomposition in the supplement. It seems to be a good result.
> >
> > Thanks for explaining the difficulty of generating meshes from neural fields. I agree that it would be difficult get exact SDF with neural fields.
> >
> > I have read the comments of other reviewers and found no serious problems. They all seem a little concerned that each component is not technically novel.
> >
> > Thanks for the feedback. Some of my concerns have been addressed. This is a good paper and I hope it is accepted.

---

### Official Review · Reviewer_vLJX · 2022-07-11

**Rating:** 7
**Confidence:** 3
**Soundness:** 3 good
**Presentation:** 4 excellent
**Contribution:** 3 good

**Summary:**

This paper proposed GET3D, a generative model that synthesizes high-quality 3D meshes with textures.
The authors put the recent 3D shape reconstruction work [40] into a styleGAN architecture and demonstrated that this framework is capable to generate diverse 3D meshes with various textures. The experiments on multiple datasets show the proposed method is superior to the prior arts.

**Questions:**

I do not have major concerns. Just some minor questions and comments as listed in the weakness section.

Overall I think this is a high-quality paper. The idea is straightforward and the results are convincing. Though I have some minor concerns, I lean into accepting this paper.

**Limitations:**

The authors listed two major limitations in their work: 1. the evaluation was done on the synthetic data only and 2. per category training. The authors also discussed the potential social impact of their work applying on privacy or biased data. All these sound sufficient to me.


**Strengths And Weaknesses:**

The writing is clear and the idea is easy to follow.

The authors closely followed the most recent 3D mesh reconstruction work nvdiffrec [40] and developed further to incorporate the 3D reconstruction pipeline into a GAN. I am glad to see that without many bells and whistles, the proposed GET3D works pretty well on multiple datasets.

The experiments are sufficient overall, though I do have some minor comments.

Weakness:
1. It is great that the generated 3D textured meshes can be directly used in the production tools like Blender. I wonder how the texture is represented in the final output as I don't think these tools can utilize texture fields? Is it easy to get the UV map from the proposed model? Or does "texture" only mean "colored vertices"?
2. Based on Table 1, [46] actually is closer to what the authors proposed in this paper (without the texture part though). Why [46] is excluded from the comparison?
3. What is the illumination model in the differentiable rendering part? If I understand correctly, it is a Phone model with only 1.0 ambient lighting.
4. Though in L255, the authors claimed that "hence the subdivision cannot provide further improvements", in Table 2, subdiv actually hurts the performance seriously. Any idea?
5. Shape interpolation is impressive. It would be great to see the texture interpolation as well (though the title of Fig 6 is shape interpolation, the texture has also changed...why...).

---

> ### Author Response · Authors · 2022-08-02
> **Thank you for your valuable comments!**
>
> We would like to thank the reviewer for the positive and detailed feedback and for perceiving our results as convincing and sufficient. We are also happy that the paper came across as well-written and easy to follow.
>
> Below, we reply to individual comments and questions raised by the reviewer:
>
> 1. **UV Map:**
> This is a great point and we should have made it more clear that, with *texture*, we don’t mean colored vertices, but rather an actual 2D texture map. As briefly described in footnote 1 (Page 7), the UV map can easily be obtained from our representation in the following manner: i) we first use the xatlas [54] package to obtain the texture coordinates of the extracted mesh from the DMTet layer, ii) using these texture coordinates, we can unwarp the 3D mesh onto a plane and obtain the corresponding 3D location on the mesh surface for any position on the 2D plane, we then iii) discretize the 2D plane into an image, and for each pixel, we query the texture field using the corresponding 3D location to obtain the RGB color. The obtained image represents the UV texture map. Note that this process only happens during inference when we export the generated textured mesh, and not during training.
>
>
> 2. **Comparison to [46]:**
> We mainly focused on comparing methods that are capable of generating shapes with varying topologies, as we see this as a crucial component for high-quality 3D asset generation. While [46] can indeed generate textured mesh, its mesh is obtained by deforming a sphere, which means that the topology is fixed to genus 0 and it is thus hard for [46] to represent complex objects (with higher genus surfaces), like motorbikes. Qualitatively speaking, the results in our paper are significantly better than the results reported in [46].
>
> 3. **Illumination model:** We provide details of our illumination model in Supplementary material (Section D). Specifically, we adopt the Disney BRDF and represent the lighting with 32 Spherical Gaussian lobes. Our illumination model can hence represent more complex lighting effects than the Phong model.
>
> 4. **Volume subdivision:** We found subdivision only improves our results on categories with thin structures (e.g. chair, motorbike) and hurt performance slightly for other categories, in contrast to [51].  We hypothesize that this is related to the loss function and supervision signal. [51] use full 3D supervision, while we only have supervision on rendered masks/images. One possibility is that the rasterizer could be affected by having too many sliver triangles after subdivision, leading to numerical instability in gradient computation. We will tone down our claim and plan to investigate this problem in future work.
>
> 5. **Interpolation:** we apologize for the confusion. In Fig. 6, we interpolate the latent code for both texture and geometry. Therefore, both of them are changed during interpolation. We have fixed it in our revised version.
>
> If there are any further concerns, we would be happy to address them in further discussion. Thank you!

---

> > ### Comment · Reviewer_vLJX · 2022-08-09
> > **Interpolation**
> >
> > Thank you for the detailed response and the feedback is satisfactory. Fig 6 makes more sense now.
> >
> > Just wonder what the results will look like if you interpolate the shape code and texture code individually. I believe this could further improve the paper (not a must-do).

---

> > > ### Author Response · Authors · 2022-08-10
> > > **Thank you for the feedback!**
> > >
> > > We thank the reviewer for the feedback!
> > >
> > > That's really great point to interpolate the geometry code and texture code individually! We provide two such results in our revised main paper (see Sec 6.5, Fig 16 & 17). Please check it.
> > >
> > > For each figure,  at every row, we interpolate the geometry latent code while fixing the texture latent code, and at every column, we interpolate the texture latent code while fixing the geometry latent code. These results demonstrate that our model is not only capable of decently disentangling geometry and texture during generation, but also can generate meaningful interpolation for either geometry or texture.
> > >
> > > We thank the reviewer for providing this suggestion! If the reviewer has any further questions or suggestions, we are more than happy to take them. Thank you!

---

### Official Review · Reviewer_9TSe · 2022-07-11

**Rating:** 6
**Confidence:** 3
**Soundness:** 3 good
**Presentation:** 3 good
**Contribution:** 3 good

**Summary:**

This paper proposes a generative method to produce textured 3D mesh that can be directly used in general 3D rendering engines. The overall pipeline contains a geometry generator, a texture generator, a differentiable rendering module, and two discriminators. The geometry generator exploits DMTet as surface representation, and a tri-plane texture field is used for the texture generator.  Besides, the differentiable module is used to predict the 2D silhouette and 3D on-surface coordinates that are used to query the corresponding RGB color in the texture field. The two discriminators are inspired by StyleGAN for silhouette and rendered RGB images to achieve RGB-supervised training, respectively.  In general, this paper is a combination of DMTet, texture field, NVdiffrec, and StyleGAN. While the performance excell SOTA method, the novelty is limited.

**Questions:**

Please provide some discussion about the weakness

**Limitations:**

Yes

**Strengths And Weaknesses:**

The strength of this work is the generated texture mesh can be directly used in general 3D rendering engines, which is not suitable for previous works. Meanwhile, the supervision is 2D images that are more widely available than 3D geometry data.
The weakness of this work contains:
1. Although this work only requires 2D images as supervision, all experiments are trained with rendered RGBs with known camera parameters and light variation. The application for in-the-wild RGB images needs more discussion. Hence the application scenario of this method is limited.
2. The author claims their method can generate high-quality 3D textures meshes, but local details (e.g. face) of the human character are missed (as in Fig.1, Fig.5). A qualitative comparison of local detail with previous SOTA methods (e.g. EG3D) will make this claim more valid.
3. The lack of quantitative comparison of human character.

Typo:
P7, L222, "Tbl. 2".
Supp. P3, L101, "Fret"

---

> ### Author Response · Authors · 2022-08-02
> **Thank you for your valuable comments!**
>
> We would like to thank the reviewer for the positive feedback and for appreciating that our generated shapes can be directly used in general 3D render engines, and the supervision in 2D images is more wildly available than 3D.
>
> Below, we reply to individual questions and comments raised by the reviewer:
>
> **1. Supervision & Applications:**
>
> First, we hope to clarify that our model only assumes known lighting **distribution** for the training dataset and **NOT** the GT lighting of each image (See Supplementary Material Section D).  For the camera pose of each image, we demonstrate that our model works decently without knowing the exact camera poses of each image in Section C.3.2 of the Supplementary Material. Furthermore, as mentioned in our general reply to all reviewers, we conducted an additional experiment in which we used noisy camera poses during training and achieved qualitatively almost indistinguishable results (Section 6.1). We also conduct an additional experiment on training GET3D on StyleGAN generated realistic dataset following GANverse3D [55] and achieve decent 3D generation with rough cameras and imperfect silhouettes. These experiments demonstrate the robustness of our method and we believe that our model makes an important step towards training 3D generative models purely from real-world images.
>
>
> We also respectfully disagree with the reviewer on the conclusion that our application scenario is limited. In our work, we have already shown two applications of our generative model: i) learning surface materials, which enables us to relight the generated 3D assets, and ii)  text-driven 3D shape generation. The network architecture of GET3D also lends itself nicely to other applications such as: single image 3D reconstruction, shape completion, shape editing, etc. which we plan to revisit in our future work.
>
>
> **2. Local details on human character generation:**
>
> We wish to point out that it is not fair to us to only focus on faces and compare our human character generation results to results reported in EG3D [8]. In our experiment (Fig 1 & 5), our model is trained to generate the **whole** human body including clothes, shoes, legs, hands, etc, while EG3D **only** generates human faces without other parts of the body. We hope that the reviewer agrees that the first is a much more challenging setting. Nevertheless, we have added an experiment in which we tried to make the comparison fair, by training EG3D on the same dataset as GET3D and tasking it to generate the whole human body (see Section 6.3). Our model performs significantly better than EG3D in terms of generating more accurate and diverse 3D shapes with textures.  Similarly, we observe that  GET3D consistently generates higher quality and more diverse 3D shapes than EG3D [8] on other datasets/categories (see Table 2 and Fig. 3).
>
>
>
> **3. Typos:**
>
> Thanks for pointing out typos, we have now fixed them in our revised version.
>
> If our reply successfully addresses the concerns of the reviewer, we would like to kindly ask the reviewer to consider raising their score accordingly. If there are any further concerns, we would be happy to address them in further discussion. Thank you!

---

> > ### Comment · Area_Chair_R67Y · 2022-08-09
> > **reminder for discussion**
> >
> > Dear reviewr,
> >
> > Thank you for providing valuable comments. The authors have provided detailed responses to your comments. Has the response addressed your major concerns?
> >
> > I would appreciate it a lot if you could reply to the authors’ responses soon as the deadline is approaching (Tues, Aug 9).
> >
> > Best,
> >
> > ACs

---

### Official Review · Reviewer_n6SS · 2022-07-11

**Rating:** 8
**Confidence:** 4
**Soundness:** 3 good
**Presentation:** 3 good
**Contribution:** 3 good

**Summary:**

This paper proposes a 3D textured mesh generator trained using 2D images, known camera distributions and object silhouettes. The generator is composed of a geometry generator, which generates the SDF value and vertice deformations to a tetrahedron field. The mesh is then extracted using differentiable marching tets. The texture generator generates a texture field, which predicts the reflectance of a point in space using its coordinates and the latent codes. By differentialbly rendering them into images and sillhouettes, image-based discriminator is used to provide the signal for training.
In addition, the authors also presented two variants, first is generating the material maps instead of reflectance, where the generated objects can be relit. Second, using a clip discriminator, the generator can be tailored to perform the task of text-to-textured-mesh.

**Questions:**

In general I find this paper above the acceptance bar. The following questions are intended to elicit more insights and details of the proposed methods:

1. How robust is the proposed method to distribution shifts of camera poses? Since it samples from the distribution of camera poses, does it really matter to have access to the exact camera pose, which seems to be a limitation of the method to apply to real images?

2. In addition to camera pose, there's also the matter of camera intrinsics. For real images, those are usually unknow and varying as well. Will the method robust enough to handle that?

3. How delicate is the system to train successfully? Though GANs are hard to tune overall, it would be nice to have some insights on how easy it is for others to re-train the system on different categories.

**Limitations:**

The authors provided adequate discussions on the limiation of the method.

**Strengths And Weaknesses:**

Strengths:
Originality:

Even though the core compnent of the system is based on prior works(nvdffrast, tet marching), the system itself is novel and nontrivial to setup. As far as I know, the proposed system is the first of its kind to generate high-quality textured mesh.

Significance:

The proposed system tackels multiple drawbacks of previous results; it uses mesh as the final geometric representation, but also not constrained by topology nor utilizes a template mesh. The quality of the result is also appealing, making it a promising candidate for generating assets for graphics applications.

Quality:

This paper is of high quality. In addition to technical details, this paper presents multiple experiments with two variants of the method to support the validity of their design choices and the potential of the method.

Clarity:

This paper is well written and easy to follow.

---

> ### Author Response · Authors · 2022-08-02
> **Thank you for your valuable comments!**
>
> We would like to thank the reviewer for the detailed feedback and for appreciating the quality of our generated shapes. We are glad that the reviewer finds our paper to be high-quality, well-written, and easy to follow.
>
> In the following, we reply to individual questions and comments raised by the reviewer:
>
>
> **1. Robustness to distribution shifts of camera poses:**
>
> Our method does not require an exact camera pose for each training image. The supplementary material includes an ablation study (Section C.3.2) that compares using exact camera poses for each image in the training data with sampling from the distribution of camera poses. There is almost no difference in the quantitative results, indicating that the exact camera pose is not necessary. In fact, the main reason to condition our discriminator on the camera pose is that it simplifies the computation of the 3D evaluation metrics on ShapeNet where all the shapes are represented in their canonical pose.
>
>  To further demonstrate the robustness of training with noisy camera poses, we conduct an additional experiment by randomly perturbing the camera with Gaussian noise as described in the general response to all reviewers. Quantitative and qualitative results of this experiment are provided in the revised version (see Section 6.1 in the main paper). The qualitative results are close to the model trained with ground truth camera poses, demonstrating the robustness to a moderate level of distribution shift of camera poses.
>
>
> **2. Different intrinsics:**
>
> This is a very good point and we agree with the reviewer that the real-world images might be captured with different intrinsics. However, one approach that can help mitigate this problem is to learn a “virtual camera” by assuming all the images are captured using a camera with the same intrinsics but vary in camera poses. This idea has been explored in [a] and EG3D [8], and could easily be plugged into our method.
>
> **3. Training stability:**
>
> Thanks to the R1 regularization [35] applied to the discriminator (Eq. 1) the GAN in GET3D is quite robust in our experiments. In our work, we have experimented with 6 different categories and they all work well without any changes to model architectures. In fact, the only hyperparameter we changed is $\lambda$ for R1 regularization (Car: 40, Animal: 40, Motorbike: 80, Human 80, House: 200, Chair: 3200). To enable fair comparisons and easier reproduction of our results, we will also release the source code if the paper is accepted.
>
>
> [a] Omni3D: A Large Benchmark and Model for 3D Object Detection in the Wild
>
> Garrick Brazil, Julian Straub, Nikhila Ravi, Justin Johnson, Georgia Gkioxari

---

> > ### Comment · Reviewer_n6SS · 2022-08-07
> > **Thanks for the response.**
> >
> > Thank you for the updated materials and response. It is assuring to see the results with noisy camera poses, as well as the hyperparameter specs for different categories.
> >
> > On the intrinsics though, converting images with different focal length, even without lens distortions, to images with the same intrinsics but different 'pose' is actually only an approxiamtion, where the depth ranges of the object should be limited. It's probably less perceptile to human when the object is cars and bikes, but for faces, which we are super good at telling minor differences, those effects are quite visible[1].
> >
> > [1]http://www.danvojtech.cz/blog/2016/07/amazing-how-focal-length-affect-shape-of-the-face/
> >
> > This is not an objection to the response, just merely a discussion on the matter of intrinsics. I don't think this is a huge issue for this paper though, but it is harder than it seems when the camera intrinsics are unkown.

---

> > > ### Author Response · Authors · 2022-08-08
> > > **Thanks for the quick reply!**
> > >
> > > We thank the reviewer for appreciating our paper and additional experiments!
> > >
> > > Regarding the unknown camera intrinsics, we thank the reviewer for pointing out the very interesting adverse effects of approximating the focal length change through the change in camera pose. It is indeed a very hard task to learn intrinsics, extrinsics, and 3D geometry from image data since they are highly entangled. Assuming the same intrinsics is a crude approximation that might help reduce the problem (though we agree it doesn't solve it). To tackle this highly-entangled problem, it might be better to utilize some priors from 3D (e.g. human face has a particular distribution of geometry) to regularize the solution space. In such cases, one would ideally have a dataset in which the same person would be captured with different cameras (intrinsics & extrinsics), making the disentanglement of individual factors easier.

---

### Author Response · Authors · 2022-08-02
**General reply to all the reviewers with additional experiments**

We thank all the reviewers for their insightful reviews. Before addressing the specific questions in the individual replies, we provide a short description of the additional experiments that we have added at the end of the original submission (marked with blue font). Specifically, we aim to address the common concerns regarding the camera poses, 2D silhouettes, real images and comparison with EG3D on human characters. To this end, we provide four additional experiments:

1. **Noisy camera poses:**
In this experiment, we randomly perturb the camera poses with Gaussian noise during training on the ShapeNet cars category. Adding camera noise harms the FID, while qualitatively we don’t observe significant differences in the results of our original model. This result demonstrates the robustness of our method to a moderate level of noise in the camera poses.

2. **Imperfect 2D silhouettes:**
To imitate how one might obtain 2D segmentation masks in real-world applications, we replace the ground truth silhouettes of the ShapeNet cars category with the ones obtained from Detectron2 using a pretrained PointRend model. We observe a drop in the FID scores, while qualitatively the results are hard to distinguish from the ones generated by our original model. This shows that GET3D is not overly sensitive to imperfect masks obtained from a pre-trained 2D segmentation network. We would also like to note that GET3D can/will benefit from future advances in 2D image segmentation, while methods that require full 3D supervision are hard to reduce their supervision cost.

3. **“Real” Images:**
Since many real images lack camera poses, we follow GANverse3D [56] and utilize pretrained 2D StyleGAN to generate realistic images with good camera initializations and 2D silhouettes. Our method performs decently well in this dataset and generates reasonable 3D textured meshes, demonstrating the potential applicability to real-world data.

4. **Comparison with EG3D on Human characters:**
Following reviewer 9TSe’s suggestions, we additionally train EG3D on the human character dataset and compare it with our method. In this setting, the models are tasked with generating the entire human body including the clothes. GET3D significantly outperforms EG3D in terms of 3D shape synthesis quality (FID-3D) and achieves comparable performance on multiview 2D image generation (FID-Ori).

It was demonstrated in the original submission that GET3D can generate 3D textured shapes of excellent quality and can be applied to several applications, including material decomposition and text-guided shape generation. These additional results should further support its position as a versatile SoTA generative model of high-quality textured 3D shapes with strong robustness to imperfect inputs.

---

### Author Response · Authors · 2022-08-08
**Thanks for the reviews!**

Dear Reviewers,


Thank you again for your thorough reviews. We would like to kindly remind you that the author-reviewer discussion period ends on Aug. 9 (in 2 days). We would appreciate it if you could have a look at our replies and let us know if they address your questions satisfactorily or if there are any further follow-up questions.


We would appreciate any additional feedback and would be happy to discuss it further.


Thank you very much,

The authors

---

### Meta-Review · Area_Chair_R67Y · 2022-08-28

**Recommendation:** Accept
**Confidence:** Certain

**Metareview:**


The paper proposes a generative model for synthesizing textured 3D meshes given only a collection of 2D images. The paper has received overwhelmingly positive reviews. Many reviewers find the idea interesting, the paper well-written, the results compelling, and the experiments comprehensive. The rebuttal further addressed the concerns such as camera poses and missing comparisons. The AC agreed with the reviewers’ consensus and recommended accepting the paper.

**Award:**

No

---

### Decision · Program_Chairs · 2022-09-14

Accept